# GaussianMarker: Uncertainty-Aware Copyright Protection of 3D Gaussian Splatting

**Xiufeng Huang[1,2], Ruiqi Li[1], Yiu-ming Cheung[1], Ka Chun Cheung[2],**
**Simon See[2], Renjie Wan[1]\***
[1] Department of Computer Science, Hong Kong Baptist University
[2] NVIDIA AI Technology Center
xiufenghuang@life.hkbu.edu.hk, {csrqli, ymc}@comp.hkbu.edu.hk
{chcheung, ssee}@nvidia.com, renjiewan@hkbu.edu.hk

## Abstract

3D Gaussian Splatting (3DGS) has become a crucial method for acquiring 3D assets. To protect the copyright of these assets, digital watermarking techniques can be applied to embed ownership information discreetly within 3DGS models. However, existing watermarking methods for meshes, point clouds, and implicit radiance fields cannot be directly applied to 3DGS models, as 3DGS models use explicit 3D Gaussians with distinct structures and do not rely on neural networks. Naively embedding the watermark on a pre-trained 3DGS can cause obvious distortion in rendered images. In our work, we propose an uncertainty-based method that constrains the perturbation of model parameters to achieve invisible watermarking for 3DGS. At the message decoding stage, the copyright messages can be reliably extracted from both 3D Gaussians and 2D rendered images even under various forms of 3D and 2D distortions. We conduct extensive experiments on the Blender, LLFF, and MipNeRF-360 datasets to validate the effectiveness of our proposed method, demonstrating *state-of-the-art* performance on both message decoding accuracy and view synthesis quality. Project page: https://kevinhuangxf.github.io/GaussianMarker.

## 1 Introduction

3DGS [1] has introduced a new category of 3D assets that can be readily created and extensively distributed online [2]. However, the ownership of these created 3D assets can be vulnerable if malicious users distribute and manipulate the 3DGS without authorization. *How can we effectively protect the ownership of those created 3DGS models?*

3DGS represents the scene via 3D Gaussian parameters, which can be standardized into point cloud formats. Such formats can be easily shared and show strong compatibility with the mainstream 3D assets processing pipeline [3]. However, unauthorized users can exploit this convenience to distribute 3DGS models and maliciously alter the 3D Gaussian parameters. These unauthorized 3DGS models can then be easily used to produce 2D images. Since ownership of 3DGS models can be compromised through unauthorized manipulations of 3D Gaussian parameters and 2D images, an effective ownership solution should enable owners to assert their rights over both the 3D Gaussian parameters and the corresponding 2D images.

Similar to copyright protection for digital assets such as videos and images, protecting copyright for 3DGS models can be achieved via digital watermarking. Aligned with the established principles in digital watermarking [4, 5], effective copyright protection methods for 3DGS models should satisfy

---

\*Corresponding author.

38th Conference on Neural Information Processing Systems (NeurIPS 2024).

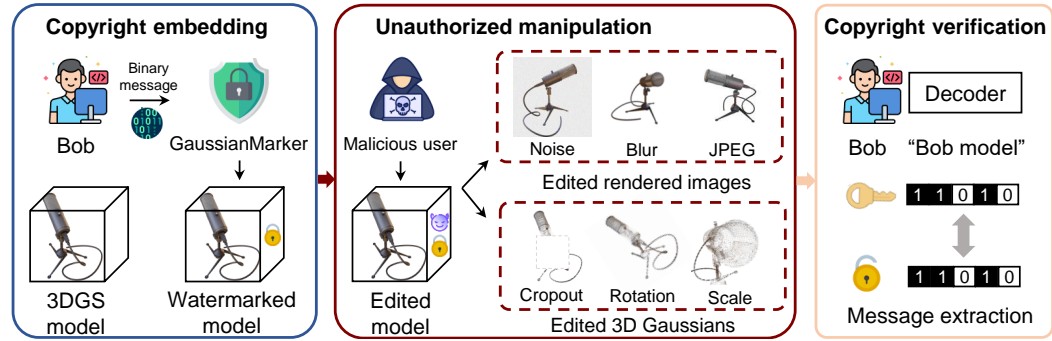

Figure 1: Our proposed scenario for copyright protection over the 3DGS assets. Once users have created 3DGS assets, they can apply our proposed 3DGS watermarking method to create watermarked 3DGS models. If unauthorized users maliciously apply 3D editing or different volume splatting settings on the watermarked 3DGS model, the 3DGS model owners can reliably retrieve the copyright message from the altered 3D Gaussian parameters or rendered 2D images to verify ownership.

two key standards. First, they should maintain **invisibility**, ensuring that the embedded copyright messages do not cause significant distortion in both 3D Gaussian parameters and the rendered 2D images. Second, they should exhibit **robustness**, enabling reliable extraction of the copyright messages even under various 2D or 3D distortions.

Although several methods [4, 6] have been investigated to protect the copyright of radiance fields, these methods are specifically designed for Neural Radiance Field (NeRF) [7], a framework known for its implicit property. For example, CopyRNeRF [4] embeds copyright messages via multilayer perceptrons (MLPs) into the implicit neural parameters in NeRF [7] and extracts the copyright messages from rendered 2D images. However, embedding messages into 3D Gaussian parameters via MLPs can easily undermine the 3D Gaussian positions and lead to noticeable geometry distortions in the rendered images, thereby degrading the **invisibility**. Additionally, since current copyright solutions for NeRF can only extract messages from rendered images, 3DGS model owners lack approaches to directly extract ownership messages from the 3D Gaussian parameters. This hinders the direct assertion of ownership over 3DGS model, thereby undermining the **robustness**.

Rather than directly embedding the copyright messages into the 3D Gaussian parameters, we propose an uncertainty-aware watermarking method to optimize the embedded copyright messages. We apply Laplace approximation to estimate the uncertainty [8] in the radiance fields for determining how large we can add perturbations to different 3D Gaussian parameters. From a Bayesian inference perspective [9], 3D Gaussian parameters with high uncertainty can tolerate larger perturbations. Thus, we keep the original 3D Gaussian parameters unchanged and densify 3D Gaussian parameters with high uncertainty. These newly densified 3D Gaussians are regarded as the perturbations for embedding copyright messages. Such perturbations can be transmitted into rendered 2D images with unperceivable distortion, which ensures **invisibility**.

To ensure robust copyright message extraction on both 3D Gaussian parameters and rendered 2D images, we utilize both 3D and 2D message decoders. The 3D message decoder extracts the copyright messages on the 3D Gaussian parameters based on a PointNet [10] architecture. The 2D message decoder based on the image watermarking method HiDDeN [11] extracts the copyright messages on rendered 2D images. We incorporate 3D and 2D distortion layers into our training process to ensure **robustness** of copyright message extraction against various malicious manipulations. The 3D distortion layer is designed to defend against malicious 3D editing, such as noise, translation, rotation, and cropping. Meanwhile, the 2D distortion layer is designed to withstand significant degradation in the rendered images, such as noise, JPEG compression, scaling, and blurring.

Our whole framework is shown in Figure 2, we estimate the uncertainty for the 3DGS model to add perturbations to different 3D Gaussian parameters. Then, we keep the original 3D Gaussian parameters unchanged and densify 3D Gaussian parameters with high uncertainty. These newly densified 3D Gaussians are regarded as the perturbations for embedding copyright messages and can be verified via 3D Gaussian parameters and 2D images. Our contribution can be summarized as follows:

- A novel method to help claim the ownership of 3D Gaussian Splatting models.

- A uncertainty-aware message embedding strategy to incorporate 3D perturbations into selected 3D Gaussian parameters to achieve invisibility.

- The copyright messages can be extracted from both 3D Gaussian parameters and rendered 2D images, showing robustness to different 3D and 2D distortions.

## 2 Related works

**3D Gaussian Splatting.** 3DGS has been rapidly adopted across multiple domains and has demonstrated remarkable results. Unlike NeRF [7] and its variants [12–14] reply on the implicit neural representation (INR) to reconstruction the 3D scene, 3DGS [1] has an explicit point cloud structure and has been expanded to various developments and applications. Mip-Splatting [15] utilizes a 3D smoothing filter and a 2D Mip filter to address frequency constraints for effective anti-aliasing. Dynamic 3DGS [16] represents the dynamic motion of the scene by processing the center location and rotation of each Gaussian over time, which enables dense non-rigid 6-DOF tracking of the entire scene. SuGar [17] reconstructs the mesh surface with a regularization term for the Gaussian Splatting optimization to promote alignment of the Gaussians with the scene's surface. 3DGS avatar [18] is a brand-new way to create digital humans compared with traditional methods based on 3D human meshes such as SMPL [19]. With the rapid development of point-based 3D Gaussian rendering, it is necessary to develop an efficient copyright protection method for 3DGS [1] models.

**2D digital watermarking.** Traditional 2D watermarking methods typically embed information in the least significant bits (LSB) of image pixels [20]. Other advanced methods encode information into the frequency domains based on the Discrete Wavelet Transform (DWT) and Singular Value Decomposition (SVD) [21, 22]. Deep-learning [23–26] has made significant progress in image watermarking [27–33]. HiDDeN [11] is one of the first deep image watermarking methods that outperformed traditional methods. RedMark [5] introduces scalable residual connections for embedding binary images in any transform domain. Robustness is a critical requirement for watermarking, ensuring resilience against various distortions and even adversarial attacks [34, 35]. Deep-learning-based watermarking methods have emerged as a crucial component in video copyright protection [5, 28, 36], such as RivaGAN [36], which utilizes an attention-based mechanism for embedding hidden messages in videos. However, the 2D digital watermarking methods for images or videos can differ significantly from 3D digital watermarking methods for explicit 3D models.

**3D digital watermarking.** Most 3D digital watermarking approaches are designed for explicit 3D models [37–41]. For example, Deep 3D-to-2D [41] can embed messages in 3D meshes [38, 42] and retrieve them from 2D rendered views [43]. Recently, several 3D digital watermarking approaches [4, 6, 44, 45] have emerged for NeRF [7] to watermark the implicit neural representation (INR) and extract the hidden information from the rendered images. CopyRNeRF [4] generates watermarked color representations to ensure the invisibility of hidden copyright messages. StegaNeRF [6] designs an optimization framework for steganographic information embedding in NeRF renderings. However, both explicit 3D watermarking [37, 38, 41] and NeRF watermarking approaches [4, 6] are not applicable for 3DGS to simultaneously protect the explicit 3D Gaussians and the 2D rendered images. This motivates us to develop digital watermarking for 3DGS models.

## 3 Preliminary of 3D Gaussian Splatting

Starting from a sparse set of Structure-from-Motion (SfM) [46] points, the goal of 3DGS [1] is to optimize a scene representation that enables high-quality novel view synthesis. The scene is modeled as a collection of 3D Gaussians:

$$G(x) = e^{-\frac{1}{2}(x-\mu)^T \Sigma^{-1}(x-\mu)}, \tag{1}$$

where $x$ is any positions in the 3D scene, $\mu$ is the 3D Gaussian center position, and $\Sigma$ is the 3D Gaussian covariance matrix. By utilizing a scaling matrix $S$ and rotation matrix $R$, we can determine the corresponding $\Sigma = RSS^T R^T$ and ensure $\Sigma$ is positive semi-definite. The 3D Gaussians need to be further projected to 2D Gaussians for rendering by volume splatting [47] method. During rendering, 3DGS follows a typical neural point-based approach [48] to compute the color $C$ of a

pixel by blending $\mathcal{N}$ depth ordered points:

$$C = \sum_{i \in \mathcal{N}} c_i \alpha_i \prod_{j=1}^{i-1} \left(1 - \alpha_j\right), \qquad (2)$$

where $c_i$ is the color estimated by the spherical harmonics (SH) coefficients of each Gaussian, and $\alpha_i$ is given by evaluating a 2D Gaussian with covariance $\Sigma'$ [49] multiplied with a per-point opacity. Consequently, the 3D Gaussians $\mathcal{G}$ contain parameters $\boldsymbol{\theta}$ including five different properties $\{\mu, R, S, c, \alpha\}$ to represent a 3D scene.

# 4   Proposed method

Our **scenario** is shown in Figure 1. We propose embedding copyright messages into 3D Gaussian parameters to protect the copyright of 3DGS models. These messages can be extracted from both the 3D Gaussian parameters and the rendered 2D images. The proposed uncertainty-aware watermarking method can claim ownership over both 3D and 2D assets derived from 3DGS models. As mentioned in Section 1, an effective watermarking algorithm for 3DGS models should achieve both invisibility in rendered novel views, and robustness in decoded messages, via the optimization goal of:

$$\mathcal{L} = \underbrace{d_1\{\mathbf{I}, \ \hat{\mathbf{I}}\}}_{\text{rendered view}} + \underbrace{d_2\{D[\mathbf{I}], \ \mathbf{M}\}}_{\text{message decoding}}, \qquad (3)$$

where $\mathbf{I}$ is the rendered novel view, $\hat{\mathbf{I}}$ is the ground truth image, $D$ is the message decoder and $\mathbf{M}$ is the copyright message. We can use appropriate distance metrics $d_1$ and $d_2$ to estimate and minimize the error in rendered views and decoded messages. A straightforward method could be embedding the copyright messages as perturbations into the 3D Gaussian parameters. However, directly embedding perturbations into 3D Gaussian parameters without constraints can easily undermine the position and geometry of 3D Gaussians and cause obvious distortion in the rendered images. To solve this issue, we propose an uncertainty-aware perturbation strategy to embed copyright messages, as illustrated below.

## 4.1   Uncertainty-aware 3DGS watermarking

**Estimating the uncertainty of Gaussian parameters.** To ensure the invisibility of embedded messages in both the 3D and 2D domains, we allow only a subset of the 3D Gaussian parameters where ownership messages can be embedded. Specifically, as previous works [9] have already shown that the Gaussian parameters with high uncertainty are more tolerant to external perturbations, we select parameters with high uncertainty to incorporate ownership messages. If we estimate the model parameter posterior $p(\boldsymbol{\theta}|\mathcal{D})$, where $\boldsymbol{\theta}$ is the model parameters of the 3D Gaussians model $\mathcal{G}$ and $\mathcal{D}$ is the training dataset, then the predictive distribution $p(\mathbf{I}|\mathbf{V}, \mathcal{D})$ can be computed by marginalize over the model posterior:

$$p(\mathbf{I}|\mathbf{V}, \mathcal{D}) = \int_{\boldsymbol{\theta}} p(\mathbf{I} \mid \mathbf{V}, \boldsymbol{\theta}) p(\boldsymbol{\theta}|\mathcal{D}) d\boldsymbol{\theta} = \mathbb{E}_{\boldsymbol{\theta} \sim p(\boldsymbol{\theta}|\mathcal{D})}[p(\mathbf{I} \mid \mathbf{V}, \boldsymbol{\theta})], \qquad (4)$$

where $\mathbf{I}$ is rendered image at a test view $\mathbf{V}$. In this inference integration, for a converged model, parameters $\boldsymbol{\theta}^*$ with larger uncertainty quantified by posterior variance can tolerate greater perturbation. Therefore, we densify only the parameters above an uncertainty threshold $\tau_{unc}$ to add perturbations for less impacting the rendered images. Laplace approximation provides an analytical expression for a posterior distribution in the form of a Gaussian distribution with the mean equal to the maximum a posterior (MAP) estimation $\boldsymbol{\theta} = \boldsymbol{\theta}^*$ [50], and the covariance equal to the reciprocal of observed Fisher information: $p(\boldsymbol{\theta}|\mathcal{D}) \sim \mathcal{N}(\boldsymbol{\theta}^*, \boldsymbol{\Gamma})$. Thus, the uncertainty of the 3DGS parameters can be estimated by the Hessian matrix $\mathbf{H}[\mathbf{I} \mid \mathbf{V}, \boldsymbol{\theta}^*]$ as the approximated Fisher information [51]:

$$\mathbf{H}[\mathbf{I} \mid \mathbf{V}, \boldsymbol{\theta}^*] = \nabla_{\boldsymbol{\theta}} f(\mathbf{V}; \boldsymbol{\theta}^*)^T \nabla^2_{f(\mathbf{V};\boldsymbol{\theta}^*)} H[\mathbf{I} \mid f(\mathbf{V}; \boldsymbol{\theta}^*)] \nabla_{\boldsymbol{\theta}} f(\mathbf{V}; \boldsymbol{\theta}^*), \qquad (5)$$

where $f(\mathbf{V}; \boldsymbol{\theta}^*)$ is the rendered image with the converged 3D Gaussians parameters $\boldsymbol{\theta}^*$ at view $\mathbf{V}$ and $\nabla^2_{f(\mathbf{V};\boldsymbol{\theta}^*)} H[\mathbf{I} \mid f(\mathbf{V}; \boldsymbol{\theta}^*)] = 1$ as we assume the covariance of RGB in images is equal to one [8]. Hence, the Hessian matrix can be simplified as: $\mathbf{H}[\mathbf{I} \mid \mathbf{V}, \boldsymbol{\theta}^*] = \nabla_{\boldsymbol{\theta}} f(\mathbf{V}; \boldsymbol{\theta}^*)^T \nabla_{\boldsymbol{\theta}} f(\mathbf{V}; \boldsymbol{\theta}^*)$.

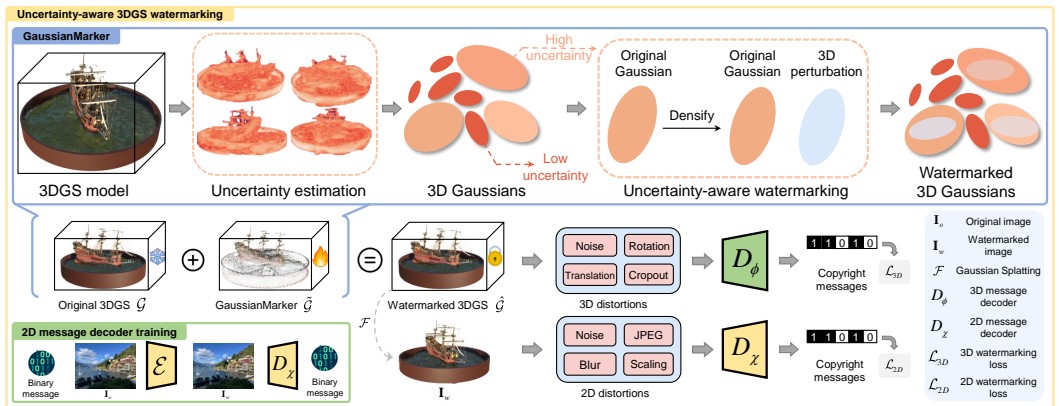

Figure 2: The overview of our proposed uncertainty-aware 3DGS watermarking. We apply uncertainty estimation to the created 3DGS model. The 3D Gaussians with high uncertainty will be densified. These new densified Gaussians will be regarded as the 3D perturbations and embedded into the original Gaussians to create watermarked 3D Gaussians. The copyright messages can be retrieved from the watermarked 3D Gaussians via the 3d message decoder under various 3D editing. The copyright messages can also be retrieved from the watermarked images via the 2D message decoder against various 2D distortions.

As Fisher Information is additive, we compute the model uncertainty $\mathbf{U}$ by summing the Hessians of model parameters across all different views in the training dataset $\mathcal{D}$:

$$\mathbf{U} = \sum_{i=1}^{N} \mathbf{H}\left[\mathbf{I} \mid \mathbf{V}, \boldsymbol{\theta}^*\right], \tag{6}$$

where $i$ is the index and $N$ is the total samples in $\mathcal{D}$, and all Gaussian parameters are used to calculate the uncertainty of the 3DGS model: $\mathbf{H}\left[\mathbf{I} \mid \mathbf{V}, \boldsymbol{\theta}^*\right] = \mathbf{H}\left[\mathbf{I} \mid \mathbf{V}, \boldsymbol{\theta}_\mu^*\right] + \mathbf{H}\left[\mathbf{I} \mid \mathbf{V}, \boldsymbol{\theta}_\mathbf{R}^*\right] + \mathbf{H}\left[\mathbf{I} \mid \mathbf{V}, \boldsymbol{\theta}_\mathbf{S}^*\right] + \mathbf{H}\left[\mathbf{I} \mid \mathbf{V}, \boldsymbol{\theta}_\mathbf{c}^*\right] + \mathbf{H}\left[\mathbf{I} \mid \mathbf{V}, \boldsymbol{\theta}_\alpha^*\right]$.

**GaussianMarker.** As shown in Figure 2, we demonstrate the overall framework of our proposed uncertainty-aware 3DGS watermarking, aka GaussianMarker. By leveraging the quantified uncertainty $\mathbf{U}$ of the created 3DGS model, we can effectively distinguish between parameters that are resilient to perturbations and those that are vulnerable. Parameters exhibiting low uncertainty are identified as highly sensitive to perturbations. Conversely, parameters characterized by high uncertainty are more tolerant to perturbations, implying that perturbations can be embedded in these areas with negligible impact on the quality of the final rendered images. By targeting 3D Gaussians with high uncertainty, we can incorporate effective 3D perturbations that remain detectable by our designated message decoders, and maintain invisibility on the 3D Gaussians and rendered images. To achieve this, we retain the integrity of the original 3D Gaussians, denoted as $\mathcal{G}$. We then densify those 3D Gaussians with high uncertainty. The new densified Gaussians are regarded as the perturbations $\tilde{\mathcal{G}}$ for copyright message embedding:

$$\tilde{\mathcal{G}} = \{g(G_i) \mid G_i \in \mathcal{G}, \mathbf{U}_i > \tau_{unc}\}, \tag{7}$$

where $g(G_i)$ is the densified Gaussian with the densify function $g(\cdot)$ on the $i^{th}$ Gaussian $G_i$ by random sampling new position $\tilde{\mu}_i$ under the distribution $\tilde{\mu}_i \sim \mathcal{N}(\mu_i, \Sigma_i)$ and other Gaussian parameters are cloned, $\mathbf{U}_i$ is the uncertainty of $G_i$, and $\tau_{unc}$ is the threshold for uncertainty. We compute the average uncertainty value of all original 3D Gaussians $\mathcal{G}$ as the default uncertainty threshold: $\tau_{unc} = \mathbf{U}/L$, where L is the total number of the 3D Gaussians in $\mathcal{G}$ [2]. We dub $\tilde{\mathcal{G}}$ as our proposed GaussianMarker for embedding the copyright messages. Similar to image watermarking methods apply 2D perturbation on the cover images, we directly embed GaussianMarker $\tilde{\mathcal{G}}$ into the original Gaussians $\mathcal{G}$ to compose the watermarked Gaussians $\hat{\mathcal{G}} = \mathcal{G} \cup \tilde{\mathcal{G}}$. Under the position sampling distribution of $\tilde{\mu}_i \sim \mathcal{N}(\mu_i, \Sigma_i)$, GaussianMarker $\tilde{\mathcal{G}}$ have a subtle geometry difference with the

---

[2]The influence of different uncertainty thresholds is further discussed in the supplementary materials.

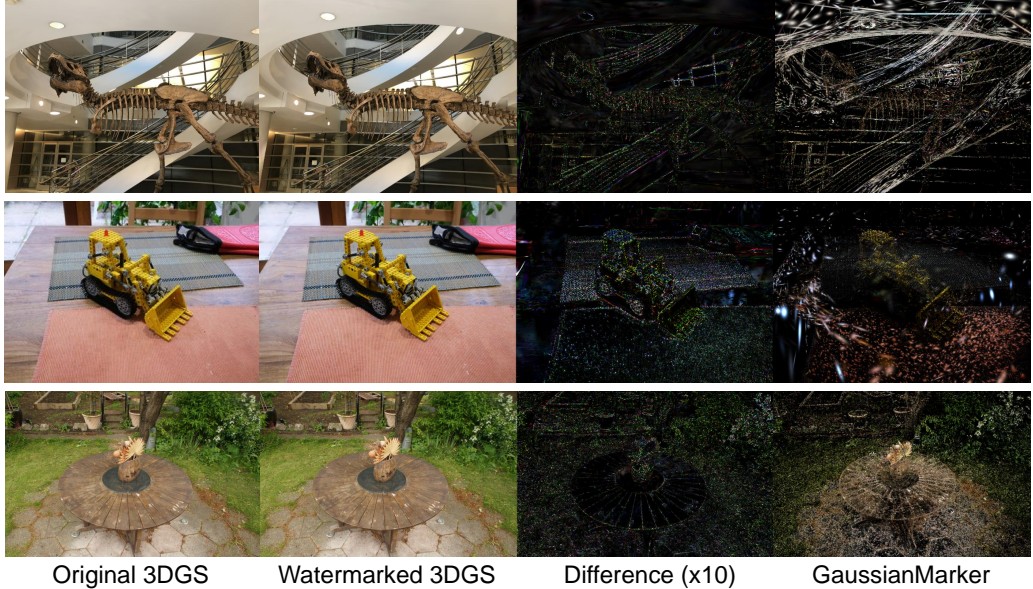

| Original 3DGS | Watermarked 3DGS | Difference (x10) | GaussianMarker |

Figure 3: Visualization of the results obtained by our proposed approach. For each row, we display the original rendered image, the watermarked rendered image, the difference ($\times 10$) between the watermarked and original rendered images, and our proposed GaussianMarker as 3D perturbations for copyright message embedding. The scalings of GaussianMarker are adjusted for better visualization. We provide more visualization examples in the supplementary material.

original Gaussians $\mathcal{G}$. Moreover, GaussianMarker $\hat{\mathcal{G}}$ can be effectively transmitted into the rendered images based on the point-based rendering in Equation (2). We optimize GaussianMarker $\hat{\mathcal{G}}$ in the Section 4.3 so that such 3D perturbations can be effectively detected by both 2D and 3D message decoders. In Figure 3, we present visualization results, which demonstrate its capability to embed copyright messages as imperceptible 3D perturbations.

## 4.2 Message decoders

**2D message decoder for rendered images.** We select the classical HiDDeN [11] as the 2D message decoder $D_\chi$ to retrieve copyright messages from the rendered 2D images. The HiDDeN [11] encoder takes a cover image $x_o$ and a binary message $\mathbf{M}$ with length $N_b$ as the inputs. The HiDDeN [11] decoder outputs a residual image of the same size as 2D perturbation applied on the original image to produce a watermarked image $x_w = x_o + \delta$. Specifically, we first pre-train HiDDeN [11] encoder and decoder to obtain a comprehensive understanding of image watermark embedding and extraction processes. During training, similar to the settings in HiDDeN [11] for the robustness at the image level, we apply several types of 2D distortions including Gaussian noise, random rotation, random cropping, and JPEG compression. After training, the HiDDeN [11] encoder and the adversarial network are discarded. We then use this pre-trained HiDDeN decoder to optimize our GaussianMarker $\tilde{\mathcal{G}}$ mentioned in Section 4.3.

**3D message decoder for 3D Gaussians.** The inherent complexity of the implicit neural representation in NeRF [7] presents significant challenges for copyright message extraction directly from the neural network parameters. On the contrary, 3DGS [1] represents the 3D scene by 3D Gaussian parameters with an explicit geometry. Traditional 3D watermarking embeds and retrieves messages from 3D meshes. The deep learning-based 3D watermarking methods utilize networks such as PointNet [10] to enhance the message embedding process in 3D meshes[41]. Although 3D Gaussians have different geometrical representations from the 3D meshes, the PointNet [10] architectures can be easily adapted to 3D Gaussians by regarding the 3D Gaussian mean $\mu$ as the point position and other parameters as the associated point features. Thus, we adopt the PointNet [10] as our 3D message decoder $D_\psi$ to retrieve copyright messages from the watermarked 3D Gaussians $\hat{\mathcal{G}}$. In specific, we randomly sample a subset Gaussians (from $10k$ to $50k$) from the watermarked Gaissians $\hat{\mathcal{G}}$ and treat these

selected Gaussians as watermarked points $\mathbf{P}_w$. A PointNet-like 3D message decoder $D_\psi$ is used to decode the copyright message $\hat{\mathbf{M}}$ from these watermarked points as $\hat{\mathbf{M}} = D_\psi(\mathbf{P}_w)$. Similarly, we also randomly sample a same number subset Gaussians on the original Gaussians $\mathcal{G}$ and treat these selected Gaussians as original points $\mathbf{P}_o$. A PointNet-like discriminator $D_\xi$ is used to distinguish the original points $\mathbf{P}_o$ and the watermarked points $\mathbf{P}_w$. The 3D message decoder $D_\psi$ uses a PointNet [10] architecture, with a modified fully connected layer to predict the copyright messages. The 3D discriminator $D_\xi$ also uses a PointNet [10] architecture with a fully connected output layer for binary classification.

## 4.3 Optimization

Our optimization contains two phases. In the first phase, we distill the watermarking knowledge from the 2D message decoder to the embedded GaussianMarker $\tilde{\mathcal{G}}$ for predicting copyright messages on rendered 2D images. In the second phase, we optimize the 3D message decoder with the watermarked Gaussians $\hat{\mathcal{G}}$ for predicting copyright messages on 3D Gaussian parameters.

**Distilling watermarking knowledge.** We keep the original 3D Gaussians $\mathcal{G}$ unchanged, and optimize the embedded GaussianMarker $\tilde{\mathcal{G}}$ via teacher-student knowledge distillation. As discussed in [52], the pre-trained feature from 2D space can be distilled to the 3D space. Thus, we use the pre-trained 2D message decoder $D_\chi$ as the teacher network to distill watermarking knowledge from the 2D perturbation on images to the 3D perturbation in GaussianMarker $\tilde{\mathcal{G}}$. During the optimization for the embedded GaussianMarker $\tilde{\mathcal{G}}$, the copyright messages can be decoded from the watermarked image $\mathbf{I}_w$ via the 2D message decoder $D_\chi$ as $\hat{\mathbf{M}} = D_\chi(\mathbf{I}_w)$, where $\hat{\mathbf{M}}$ is the copyright messages decoded by $D_\chi$, and $\mathbf{I}_w$ is rendered by the watermarked Gaussians $\hat{\mathcal{G}}$. We compute binary cross entropy (BCE) between the original message $\mathbf{M}$ and the decoded message $\hat{\mathbf{M}}$ as the message loss $\mathcal{L}_{msg}$ to ensure watermarking capability:

$$\mathcal{L}_{msg} = -(\mathbf{M}\log(\hat{\mathbf{M}}) + (1 - \mathbf{M})\log(1 - \hat{\mathbf{M}})). \tag{8}$$

We also use the photometric loss $\mathcal{L}_{rec} = \|\mathbf{I}_w - \mathbf{I}_o\|_2^2$ between the watermarked image $\mathbf{I}_w$ and the original images $\mathbf{I}_o$ for multi-view consistency. We combine the $\mathcal{L}_{msg}$ and $\mathcal{L}_{rec}$ into the final 2D watermarking loss $\mathcal{L}_{2D} = \lambda_1 \mathcal{L}_{msg} + \lambda_2 \mathcal{L}_{rec}$, where $\lambda_1, \lambda_2$ are the weights for adapting the losses.

**Optimizing 3D message decoder.** Once the embedded GaussianMarker $\tilde{\mathcal{G}}$ is optimized, we proceed to train the 3D message decoder $D_\psi$ with the watermarked Gaussians $\hat{\mathcal{G}}$. To make our 3D message decoding robust to different 3D distortions, we add a 3D distortion layer $T$ during the 3D message decoder optimization. Several commonly used 3D distortions are used: 1) additive random Gaussian noise with parameter $\sigma$; 2) random axis-angle rotation with parameter $r$; 3) random translation with parameter $t$; and 4) random cropout with parameter $cr$. The copyright messages $\hat{\mathbf{M}}'$ can be decoded from the randomly selected watermarked points $\mathbf{P}_w$ from the watermarked Gaussians $\hat{\mathcal{G}}$ via the 3D message decoder $D_\psi$ as $\hat{\mathbf{M}}' = D_\phi(T(\mathbf{P}_w))$. We compute the BCE between the original message $\mathbf{M}$ and extracted message $\hat{\mathbf{M}}'$ as the 3D message loss $\mathcal{L}_{msg'} = -(\mathbf{M}\log(\hat{\mathbf{M}}') + (1 - \mathbf{M})\log(1 - \hat{\mathbf{M}}'))$. We also apply the adversarial loss $\mathcal{L}_{adv}$ to optimize the 3D discriminator $D_\xi$ for classifying the original points $\mathbf{P}_o$ and watermarked points $\mathbf{P}_w$:

$$\mathcal{L}_{adv} = \log(1 - D_\xi(T(\mathbf{P}_o))) + \log(D_\xi(T(\mathbf{P}_w))). \tag{9}$$

We combine $\mathcal{L}_{msg'}$ and $\mathcal{L}_{adv}$ into the final 3D watermarking loss $\mathcal{L}_{3D} = \lambda_1' \mathcal{L}_{msg'} + \lambda_2' \mathcal{L}_{adv}$, where $\lambda_1', \lambda_2'$ are the weights for adapting the losses.

# 5 Experiments

## 5.1 Experimental settings

**Dataset**. We use three benchmark datasets for evaluation: **Blender** [7] (8 detailed synthetic objects), **LLFF** [53] (9 real-world scenes), and **Mip-NeRF360** [54] (9 real-world scenes). For Blender [7], we directly follow the dataset splitting to use 100 viewpoints for training and 200 views for testing. For LLFF [53], we follow the dataset splitting in NeRF [7]. In general, $1/8$ images in each scene are

used for testing and others for training. For Mip-NeRF360 [54], we use a train/test split suggested by Mip-NeRF360, taking every $8^{th}$ photo for testing and others for training. All testing viewpoints are used to compute the average values during the evaluation session.

**Implementation details.** As our motivation is to protect the copyright of 3DGS that has already been created, we define our training into two stages. In the first stage, we create 3DGS models by training them on Blender [7], LLFF [53], and Mip-NeRF360 [54] datasets following standard settings [1]. We also train HiDDeN [11] to obtain the pre-trained 2D message decoder. In the second stage, we apply our proposed uncertainty-aware watermarking method to generate GaussianMarker $\tilde{\mathcal{G}}$ via Equation (7). We then freeze the original Gaussians $\mathcal{G}$ and use the pre-trained 2D message decoder to supervise the training of our GaussianMarker $\tilde{\mathcal{G}}$. We then train our 3D message decoder to retrieve the copyright message from the watermarked 3D Gaussians $\hat{\mathcal{G}}$. We apply several types of 2D and 3D distortion layers on the watermarked 2D images and 3D Gaussians to achieve robustness. We use the default optimization setting in 3DGS [1] to optimize our GaussianMarker $\tilde{\mathcal{G}}$. We use the Adam optimizer [24] to optimize the 3D message decoder $D_\psi$ and classifier $D_\xi$ with default values $\beta_1 = 0.9, \beta_2 = 0.999, \epsilon = 10^{-8}$, and a learning rate $1 \times 10^{-4}$ that decays following the exponential scheduler during optimization. We set $\lambda_1 = 10.0, \lambda_2 = 1.0$ for 2D watermarking loss $\mathcal{L}_{2D}$ and $\lambda_1' = 2.0, \lambda_2' = 1.0$ for 3D watermarking loss $\mathcal{L}_{3D}$ to adapt the training losses. The training takes 1000 (Blender, LLFF) or 2000 steps (MipNeRF360) and can finish within 20 minutes using a single NVIDIA V100 GPU.

**Baselines.** We design experiments to validate the message extraction on both rendered 2D images and 3D Gaussian parameters, demonstrating the effectiveness of our proposed method. For 2D message extraction, we compare our proposed method with four baselines for a fair comparison: 1) **CopyRNeRF**[4]: A state-of-the-art method for protecting the copyright of NeRF [7] by using watermarked color representation; 2) **HiDDeN** [11] + **3DGS** [1]: Preprocessing images with the classical image watermarking method HiDDeN [11] before the training of 3DGS [1]; 3) **3DGS with message:** Creating message embedding by MLPs and concatenating the message embedding with 3D Gaussian parameters; 4) **3DGS with fine-tuning:** Fine-tuning all of the 3D Gaussian parameters for embedding copyright messages. For 3D message extraction, since NeRF watermarking methods do not have explicit 3D parameters, we compare our methods with the 3DGS baselines, including HiDDeN + 3DGS, 3DGS with message, and 3DGS with fine-tuning.

**Evaluation methodology.** We evaluate the performance of our proposed method by comparing it with other digital watermarking baselines using the standard of capacity, invisibility, and robustness for both 2D images and 3D Gaussians. For *capacity*, we set the bit length of copyright messages to $48$ bits, aligning with the maximum length previously employed in 3D model watermarking methods [41, 4]. For *invisibility*, we evaluate the reconstruction quality with PSNR, SSIM, and LPIPS [55] for 2D images, and we evaluate geometry difference with the $\mathcal{L}_1$ norm of position difference ($\mathcal{L}_1 Diff$), and signal-to-noise ratio (SNR) for 3D Gaussian positions. For *robustness*, we evaluate whether the copyright messages in 2D images can remain consistent against various distortions, including 2D Gaussian noise, JPEG compression, scaling, and Gaussian blur. We also evaluate whether the copyright messages in 3D Gaussians can remain consistent against various 3D attacks, including 3D Gaussian noise, translation, rotation, and crop-out.

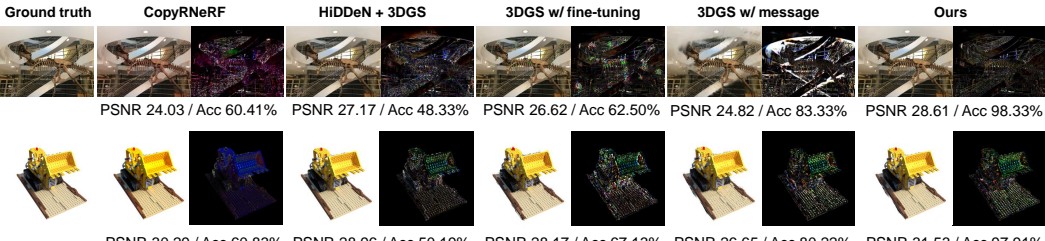

Figure 4: Comparisons between each baseline and our proposed method. We display the differences ($\times 10$) between the synthesized results and the ground truth for each method. Our proposed Gaussian-Marker demonstrates superior reconstruction quality and bit accuracy.

| Dataset | Method | PSNR/SSIM↑ | LPIPS↓ | None | Noise $(\nu = 0.1)$ | JPEG $(Q = 50)$ | Scaling $(s \leq 25\%)$ | Blur $(\xi = 0.1)$ |
|---|---|---|---|---|---|---|---|---|
| | | | | | | Bit accuracy ↑ (%) | | |
| Blender | CopyRNeRF [4] | 30.29/0.8878 | 0.0813 | 60.83 | 59.92 | 58.52 | 57.44 | 60.22 |
| | HiDDeN [11] + 3DGS [1] | 28.96/0.8812 | 0.0829 | 50.19 | 49.84 | 50.12 | 50.09 | 50.16 |
| | 3DGS [1] w/ messages | 22.65/0.8066 | 0.1584 | 80.22 | 78.66 | 75.80 | 78.08 | 79.64 |
| | 3DGS [1] w/ fine-tuning | 28.17/0.9047 | 0.0878 | 67.13 | 67.06 | 63.43 | 64.04 | 66.38 |
| | Ours | **31.53//0.9082** | **0.0759** | **97.91** | **96.93** | **91.66** | **96.17** | **97.43** |
| LLFF | CopyRNeRF [4] | 24.03/0.7747 | 0.2575 | 60.77 | 60.23 | 58.06 | 58.89 | 60.35 |
| | HiDDeN [11] + 3DGS [1] | 27.17/0.8543 | 0.1210 | 48.26 | 48.14 | 46.26 | 46.89 | 48.12 |
| | 3DGS [1] w/ messages | 24.82/0.8452 | 0.1310 | 83.33 | 82.39 | 79.17 | 81.04 | 83.18 |
| | 3DGS [1] w/ fine-tuning | 26.62/0.8566 | 0.1117 | 60.61 | 59.99 | 55.49 | 57.52 | 60.40 |
| | Ours | **28.61/0.8930** | **0.0999** | **98.33** | **97.83** | **91.45** | **95.89** | **98.23** |
| MipNeRF360 | CopyRNeRF [4] | 22.47/0.8053 | 0.4825 | 58.55 | 57.22 | 55.26 | 55.80 | 57.59 |
| | HiDDeN [11] + 3DGS [1] | 27.20/0.8151 | 0.2143 | 48.75 | 48.03 | 45.93 | 47.75 | 48.56 |
| | 3DGS [1] w/ messages | 24.84/0.7992 | 0.1705 | 77.08 | 76.75 | 74.26 | 75.54 | 77.00 |
| | 3DGS [1] w/ fine-tuning | 27.04/0.8452 | 0.1357 | 61.67 | 61.45 | 59.94 | 60.56 | 61.51 |
| | Ours | **29.16/0.8808** | **0.1197** | **97.32** | **97.01** | **90.77** | **95.32** | **97.18** |

Table 1: Reconstruction qualities and bit accuracy compared with different baselines. PSNR/SSIM and LPIPS are computed between the original and watermarked rendered images. The results are computed on the average of all examples.

| Method | Geometry difference | | None | Noise $(\sigma = 0.1)$ | Translation $(t = [0, 1000]^3)$ | Rotation $(r = \pm\pi/6)$ | Cropout $(cr = 0.1)$ |
|---|---|---|---|---|---|---|---|
| | $\mathcal{L}_1 Diff \downarrow$ | SNR↑ | | | Bit accuracy ↑ (%) | | |
| HiDDeN [11] + 3DGS [1] | 0.00912 | 40.90 | 68.20 | 67.65 | 67.32 | 66.67 | 64.24 |
| 3DGS [1] w/ messages | 0.10513 | 32.93 | 85.41 | 84.91 | 85.35 | 81.52 | 79.57 |
| 3DGS [1] w/ fine-tuning | 0.01829 | 37.24 | 69.79 | 69.70 | 68.78 | 65.88 | 64.84 |
| Ours w/ 2D decoder | **0.00003** | **43.23** | 97.85 | 57.05 | 59.07 | 53.88 | 48.23 |
| Ours w/ 3D decoder | **0.00003** | **43.23** | **100** | **99.91** | **98.95** | **95.83** | **92.70** |

Table 2: Geometry difference and bit accuracy compared with different baselines. $\mathcal{L}_1$ distance and SNR are computed between the original and watermarked 3D Gaussians. The results are computed on the average of all examples from Blender, LLFF, and MipNeRF360.

## 5.2 Experimental results

**Messages extraction with 2D images.** We compare the reconstruction qualities and bit accuracies with all baselines, and the qualitative and quantitative results are shown in Figure 4 and Table 1. CopyRNeRF [4] can limitedly extract hidden messages from the renderings and show undermined robustness to different image distortions. Although HiDDeN [11] + 3DGS [1] can achieve high reconstruction quality, it fails to extract the copyright messages from the rendered 2D images. This result is aligned with the previous method [4] and proves the message can not be transmitted from the 2D images into the 3D Gaussians. 3DGS [1] with messages directly embed copyright messages into 3D Gaussian parameters. It can retrieve the copyright message with relatively high accuracy, but the reconstruction quality is poor and shows obvious distortions in the rendered images. 3DGS [1] with fine-tuning shows better reconstruction quality, but the message extraction accuracy is limited. This is because 3D Gaussians usually contain millions of parameters, and directly fine-tuning all of the parameters without proper regularization can be less effective for message extraction. Our method can achieve both high reconstruction quality and high decoding accuracy. Even with different distortions to the rendered images, our method can still achieve high decoding accuracy to reliably safeguard the 3DGS models.

**Messages extraction with 3D Gaussians.** We evaluate the geometry differences and the bit accuracies with all 3DGS baselines, and the results are shown in Table 2. HiDDeN [11] + 3DGS [1] has small geometry difference, but it has limited message decoding accuracy. 3DGS with messages shows reasonable message decoding accuracy, but it displays high geometry differences. 3DGS with fine-tuning shows a relatively small geometry difference, but it struggles to decode the message. Our method has the smallest geometry difference, obtains accurate decoding accuracy, and shows robustness to different 3D attacks. Furthermore, 3DGS can be easily edited in the 3D space to influence the rendered 2D images. We conduct experiments to apply 3D attacks and then render the 2D images. As shown in Table 2, the 3D attacks can easily fool the 2D message decoder, while our 3D message decoder is robust to such 3D attacks and can reliably extract copyright messages.

### 5.3 Ablation study

**Perturb low-uncertainty Gaussians.** We design ablation experiments to add perturbations into 3D Gaussian parameters with low uncertainty. We show qualitative results in Figure 5, the low-uncertainty Gaussians corresponding to the fine details in the 3D scene, such as the halyard on the ship. Adding perturbations to these areas can easily make the perturbation visible, thus undermining the image quality. We evaluate the quantitative results in Table 3. Adding perturbations to 3D Gaussian parameters with low uncertainty can easily undermine the reconstruction quality and degrade the decoding accuracy. Our method preserves 3D Gaussians with low uncertainty, maintaining the geometric structure to ensure imperceptible perturbations and high message extraction accuracy.

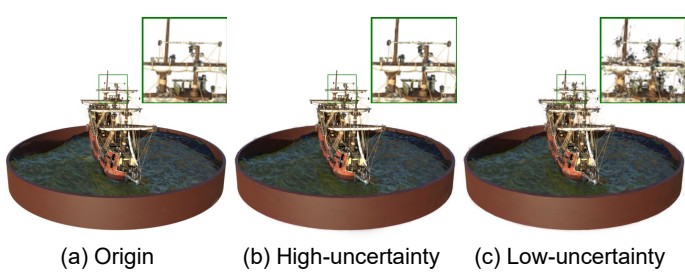

(a) Origin     (b) High-uncertainty     (c) Low-uncertainty

Figure 5: Qualitative results of applying perturbation into (b) high uncertainty Gaussians and (c) low uncertainty Gaussians.

| Metric | Low | High |
|--------|------|------|
| PSNR | 26.61 | 28.71 |
| SSIM | 0.8782 | 0.9016 |
| LPIPS | 0.0948 | 0.0763 |
| Acc | 76.56 | 97.07 |
| Noise | 75.51 | 95.83 |
| Blur | 76.27 | 96.30 |
| Resize | 73.92 | 93.22 |

Table 3: Quantitative results of adding perturbation into low-uncertainty Gaussians and high-uncertainty Gaussians.

## 6 Conclusion

In conclusion, protecting the copyright of 3D Gaussian Splatting (3DGS) assets is crucial due to their vulnerability to unauthorized distribution and manipulation. Existing methods for copyright protection in the radiance field are not directly applicable to 3DGS. Our proposed method involves using uncertainty estimation to add invisible 3D perturbations to the 3D Gaussian parameters, ensuring both invisibility and robustness. Overall, our proposed approach introduces an effective solution with a positive societal impact on the copyright protection of the 3DGS models.

**Limitations.** Our method is an effective technical solution for the copyright protection of 3DGS models. However, as we discussed before, our mechanism may still face threats from some malicious operations. More measures should be implemented for such malicious attacks beyond the technology. Furthermore, we will explore enhancing the robustness of GaussianMarker in dynamic 3DGS scenarios, via the motion transfer-based data augmentation approach, to maintain high bit accuracies while improving robustness [56] in future work.

## 7 Acknowledgement

This work was done at Renjie's Research Group at the Department of Computer Science of Hong Kong Baptist University. Renjie's Research Group is supported by the National Natural Science Foundation of China under Grant No. 62302415, Guangdong Basic and Applied Basic Research Foundation under Grant No. 2022A1515110692, 2024A1515012822, and the Blue Sky Research Fund of HKBU under Grant No. BSRF/21-22/16. This work was supported in part by the RGC Senior Research Fellow Scheme under the grant: SRFS2324-2S02.

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

# A Additional visualization

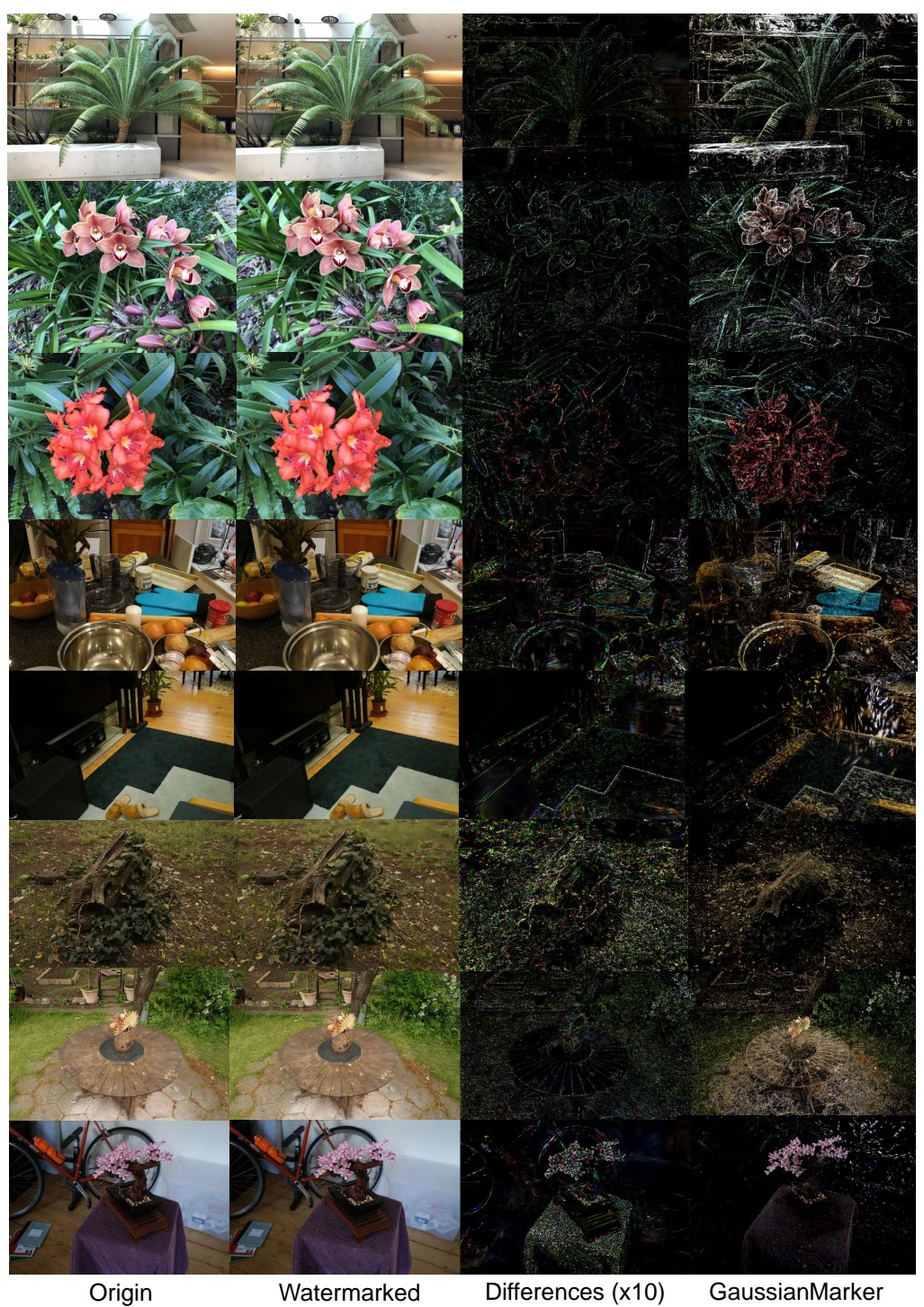

Figure 6: Visualization of our proposed method performance in LLFF and MipNeRF360 datasets. In each line, we display the original rendered image, the watermarked rendered image, the difference (×10) between the watermarked and original rendered images, and our proposed GaussianMarker as 3D perturbations for copyright message embedding. The scalings of GaussianMarker are adjusted for better visualization.

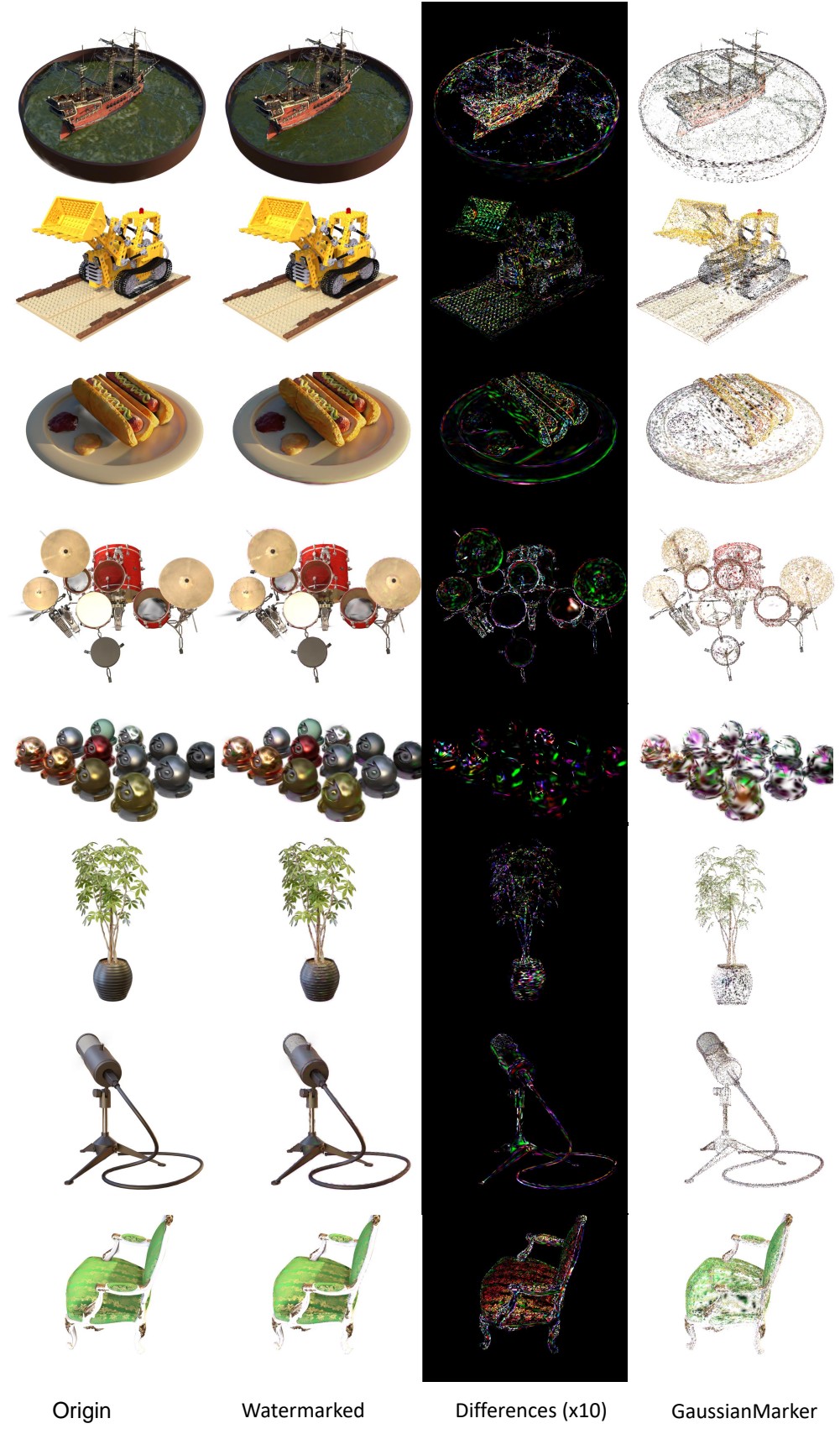

Origin    Watermarked    Differences (x10)    GaussianMarker

Figure 7: Visualization of our proposed method performance in Blender dataset.

# B  Additional quantitative results for real-world datasets

| LLFF | PSNR | SSIM | LPIPS | Acc | Noise | JPEG | Scale | Blur |
|---|---|---|---|---|---|---|---|---|
| Fern | 29.40 | 0.92 | 0.0867 | 1.0 | 1.0 | 0.8916 | 0.9791 | 1.0 |
| Fortress | 30.91 | 0.9084 | 0.1227 | 0.9375 | 0.9375 | 0.9166 | 0.9375 | 0.9375 |
| Horn | 27.46 | 0.8849 | 0.1094 | 0.9791 | 0.9791 | 0.9125 | 0.8958 | 0.9791 |
| Orchids | 25.177 | 0.8543 | 0.0853 | 1.0 | 1.0 | 0.9333 | 0.95833 | 1.0 |
| Flower | 30.09 | 0.8939 | 0.0952 | 1.0 | 1.0 | 0.9125 | 0.9791 | 1.0 |
| Trex | 28.165 | 0.9109 | 0.0734 | 1.0 | 1.0 | 0.8541 | 0.9791 | 1.0 |

Table 4: Quantative results on LLFF scenes.

| MipNeRF360 | PSNR | SSIM | LPIPS | Acc | Noise | JPEG | Scale | Blur |
|---|---|---|---|---|---|---|---|---|
| Stump | 28.95 | 0.8521 | 0.1364 | 0.9791 | 0.9791 | 0.8291 | 0.9375 | 0.9791 |
| Bicycle | 25.68 | 0.8028 | 0.1774 | 0.9583 | 0.9583 | 0.8750 | 0.9125 | 0.9583 |
| Kitchen | 29.87 | 0.8937 | 0.0946 | 0.9791 | 0.9791 | 0.8333 | 0.9583 | 0.9791 |
| Counter | 28.68 | 0.8948 | 0.1300 | 0.9583 | 0.9583 | 0.8916 | 0.9166 | 0.9583 |
| Bonsai | 30.41 | 0.9249 | 0.1002 | 0.9583 | 0.9583 | 0.9125 | 0.8958 | 0.9583 |
| Garden | 28.58 | 0.8825 | 0.0760 | 1.0 | 1.0 | 0.9125 | 0.9375 | 1.0 |
| Room | 31.97 | 0.9149 | 0.1232 | 0.9791 | 0.9791 | 0.8750 | 0.9583 | 0.9791 |

Table 5: Quantative results on MipNeRF360 scenes.

# C  The correlation between uncertainty and image watermarking

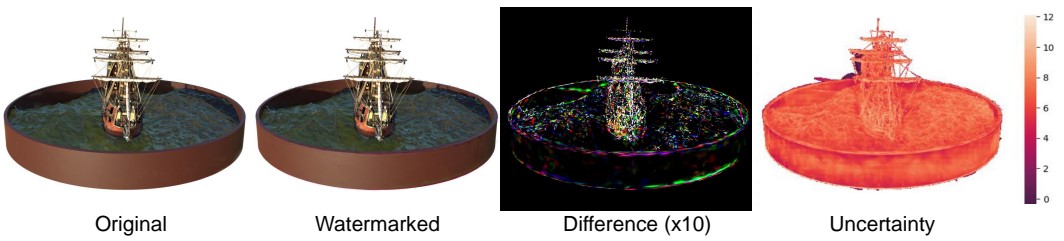

Original          Watermarked          Difference (x10)          Uncertainty

Figure 8: Uncertainty heatmap visualization.

The uncertainty estimation in the 3DGS model inherently identifies model parameters that are more robust to perturbations, making it highly suitable for the application of invisible watermarking in 3DGS. Additionally, the HiDDeN decoder [11], primarily focuses on decoding information along the boundary regions. As illustrated in Figure 8, these boundary regions exhibit high uncertainty values. This observation demonstrates the correlation between uncertainty and message embedding, highlighting how areas of high uncertainty can be leveraged for effective watermarking.

# D  The influence of uncertainty threshold

In our experiments, we set the average uncertainty value as the default threshold. We show more results to verify the influence of the uncertainty threshold. As shown Table 6, we select the ship scene in the Blender dataset to study the influence of the uncertainty threshold. A lower threshold can enhance the bit accuracy, though it slightly compromises image quality. Conversely, a higher threshold results in better image quality and a more lightweight model but also slightly compromising message decoding accuracy. However, in both situations, the compromises are moderate.

It is noteworthy that our method is also compatible with compressing the 3DGS model based on the uncertainty value, similar to how LightGaussian compresses 3DGS using importance values [57].

| Threshold | Original points | Perturbation points | PSNR | SSIM | LPIPS | ACC |
|---|---|---|---|---|---|---|
| average × 3.7 | 340k | 13k | 26.44 | 0.9521 | 0.0396 | 92.08% |
| average × 1.0 | 340k | 27k | 26.37 | 0.9498 | 0.0397 | 95.21% |
| average × 0.24 | 340k | 54k | 26.31 | 0.9490 | 0.0399 | 96.88% |
| average × 0.13 | 340k | 108k | 26.17 | 0.9483 | 0.0399 | 96.31% |

Table 6: The influence of uncertainty threshold.

This compatibility highlights that our approach can work seamlessly with existing 3DGS model compression techniques, effectively mitigating the impact of the increasing number of 3D Gaussians in our method.

## E  The geometry consistency

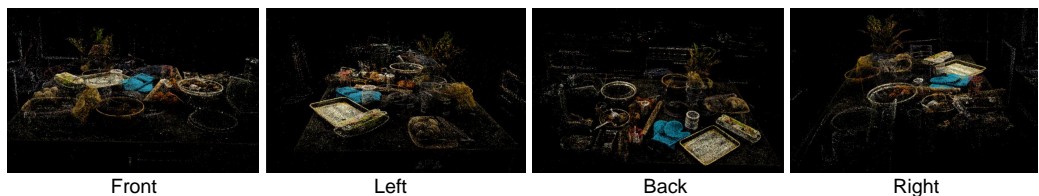

Front          Left          Back          Right

Figure 9: Visualization of our proposed GaussianMarker in MipNeRF360 **Room** scene. We select four camera angles which are never used in the training dataset as the font, left, back, right views of the scene to represent the multi-view consistency of incorporated perturbations.

Our method embeds watermarks into the 3DGS model by adding perturbations to 3D Gaussians with high uncertainty. As shown in Figure 8, these areas cover most object boundaries in the 3DGS scene. Figure 9 further illustrates geometry consistency by displaying the incorporated perturbations. These perturbations effectively cover the scene's general geometric structure. The geometry of these perturbations remains consistent across different camera angles and can be transmitted into the rendered images, which is essential for robust extraction from different viewing angles.

## F  Time analysis

| Datasets | 3DGS training | Our message embedding |
|---|---|---|
| Synthetic datasets (Blender) | 30k steps / 30mins | 1k steps / 3 mins |
| Real-world scenes (LLFF) | 30k steps / 40mins | 1k steps / 5 mins |
| Real-world scenes (MipNeRF360) | 30k steps / 45mins | 2k steps / 10 mins |

Table 7: Time analysis on different datasets.

We present a time analysis of our method's training efficiency in Table 7. Compared to the original 3DGS model training, our method requires only 1,000 to 2,000 steps within a span of 10 minutes. This demonstrates that our approach is not only efficient but also practical for watermarking 3DGS models.

