# OpenReview forum: "GaussianMarker: Uncertainty-Aware Copyright Protection of 3D Gaussian Splatting"
_NeurIPS.cc/2024/Conference — NeurIPS 2024 poster_

### Official Review · Reviewer_BKx3 · 2024-07-03

**Soundness:** 3
**Presentation:** 3
**Contribution:** 2
**Rating:** 4
**Confidence:** 5

**Summary:**

This paper proposes GaussianMarker, a novel method for embedding invisible watermarks into 3D Gaussian Splatting (3DGS) models to protect their copyright. The key idea is to use uncertainty estimation to add imperceptible perturbations to 3D Gaussian parameters with high uncertainty. The method enables extraction of copyright messages from both the 3D Gaussian parameters and rendered 2D images, and demonstrates robustness to various 3D and 2D distortions. Experiments on multiple datasets show the effectiveness of the approach in terms of message decoding accuracy and visual quality preservation.

**Strengths:**

* Timely contribution addressing copyright protection for 3D Gaussian Splatting models, an increasingly important 3D asset format
* Clever use of uncertainty estimation to guide watermark embedding in a way that preserves visual quality
* Demonstrates robustness to various 3D and 2D distortions/attacks

**Weaknesses:**

* The decoder is trained per scene, rather than being a generalizable decoder. This makes the watermarking process essentially impractical for real-world use. It's not feasible for people to store a separate watermark encoder and decoder for each scene for the vast number of Gaussians distributed across the internet. Reflecting on the logic of image watermarking, a single watermark encoder and decoder can encode and decode information for any cover image, so the sender and receiver only need to jointly possess one watermark decoder. This is a more reasonable setup.
* Experiments focus mostly on relatively simple scenes - more complex, dynamic scenes could be challenging
* The robustness to more sophisticated attacks (e.g. adversarial perturbations) is not explored
* Discussion of potential negative impacts of the technology could be expanded

**Questions:**

My primary concern with this paper stems from a fundamental physical challenge: How can a digital watermark be embedded into the 3D Gaussian Splatting (3DGS) representation of a scene in such a way that it can be reliably decoded from any viewing direction?

The volume rendering process that converts 3DGS representations into 2D images is designed to produce geometrically consistent views based on the camera pose. However, the requirement to embed and extract a watermark from arbitrary viewpoints seems to conflict with this underlying principle.

One potential resolution to this contradiction could be as follows: Rather than directly encoding the watermark itself into the 3D representation, the method might embed a geometrically consistent signal that can be detected by a trained network D. This signal could then trigger the generation or retrieval of the actual watermark (be it an image, audio, or text), which has been memorized by the detector D during the training process.

This hypothesis aligns with the paper's description of F as a detector/classifier rather than a decoder. It also explains the need for a separate classification module to guide whether the detector should produce the stored watermark data.

However, this interpretation raises several questions:

* How does the method ensure that the embedded signal remains detectable across different viewing angles and rendering conditions?
What is the information capacity of this approach, and how does it compare to traditional digital watermarking techniques?
* How robust is the embedded signal to various forms of 3D transformations or edits to the 3DGS model?
* Is there a trade-off between the strength of the embedded signal and the visual quality of the rendered images?

**Limitations:**

The authors have included a brief limitations section that acknowledges potential vulnerabilities to malicious attacks beyond their technical solution. However, this discussion could be expanded to consider more specific limitations of the approach, such as potential challenges with very complex scenes or highly dynamic content. The societal impact is briefly mentioned, focusing on the positive aspects of copyright protection. A more thorough examination of potential negative impacts or misuse scenarios would strengthen the paper.

---

> ### Author Rebuttal · Authors · 2024-08-05
>
> Dear Reviewer BKx3,
>
> We will address your comments below and in the revised paper.
>
> ### Weakness:
>
> > W1: Generalizability of decoders.
>
> We agree that the generalization ability is very important for practical use. In our designs, we employ a two-layer protection approach to achieve generalizable and specific protection. We have two decoders in 2D and 3D, respectively. The 2D decoder is **never trained per scene** for generalization ability. In our setting, it is pre-trained and can be used across images rendered from different 3DGS models. This can provide protections for **vast number of Gaussians distributed across the internet**. Our second decoder is in 3D and is trained per scene. This is the second protection wall for 3D assets with more specific protection. The 3D asset owners can train their 3D decoder per scene based on the scene property. We believe such two-layer protection with both general and specific protection can make 3D assets safer.
>
> > W2: More complex, dynamic scenes.
>
> In Table 1 of our main paper, we present evaluations of our method across three datasets, including MipNeRF360, widely regarded as the most challenging static scene dataset. To address your concerns, in this rebuttal, we further test our watermarking strategy on the Simplicits [1], a latest framework for simulating complex motions. In this place, we just need to simply modify the distortion layers in the original designs by incorporating various geometry distortions. From the **left part of rebuttal Figure 4**, our method still shows robustness in such new and complex scenarios. Besides, it also partly shows that our approach can be extended to dynamic 3DGS scenarios.
>
> > W3: Adversarial perturbations.
>
> Following your suggestions, we have designed more experiments to evaluate adversarial robustness. A malicious user can employ an adversarial attack, such as PGD, to compromise hidden messages in rendered images. As shown in the **right part of rebuttal Figure 5**, adversarial attacks can indeed reduce bit accuracy, even with minimal visual distortion (PSNR > 30). However, the defense against such adversarial attacks can be easily achieved via adversarial training by generating adversarial samples during the training for HiDDeN decoder. Under adversarial training to the message decoder for 2D images, the bit accuracy can be significantly improved, while the image quality is preserved with a PSNR value larger than 30.
>
> > W4: Potential negative impacts.
>
> Although our method demonstrates robustness against many sophisticated distortions,  additional legislative strategies should be implemented to combat such malicious attacks beyond technological measures. We will explore extending our work by integrating legislative actions to provide comprehensive copyright protection for model owners.
>
> ### Questions
>
> If our understanding is accurate, BKx3’s primary concern is whether our watermarks can maintain geometric consistency, which is essential for robust extraction from different viewing angles. Our specific designs have equipped our digital watermarks with such consistency. Our method embeds watermarks into the 3DGS model by adding perturbations to 3D Gaussians with high uncertainty.
> As shown in Figure 3 of our main paper and supplementary material, these areas cover most object boundaries in the 3DGS scene, regardless of viewing angle. **Rebuttal Figure 6** further illustrates geometry consistency by displaying the incorporated perturbations. These perturbations effectively cover the scene's general geometric structure. The geometry of these perturbations remains consistent across different camera angles and can be transmitted into the rendered images, enabling detection by the 2D message decoder.
>
> The scenarios envisioned by BKx3 can be achieved by some classical information-hiding techniques [2], which use specific strategies to hide images or even videos in the target data and can employ a specific decoder to extract such hidden data. We try to address the newly raised questions related to this envisioned scenario below：
>
> > Viewing angles, information capacity, and comparison to traditional watermarking.
>
> During training, careful selection of training and testing data guarantees that camera angles cover most scene perspectives, enabling successful watermark detection from various viewpoints. The information capacity relies on the model parameters of the decoder since this approach is conducted in an over-fitting manner. Compared with traditional watermarking, the envisioned method can decode different types of watermark data, but it relies on a specific decoder.
>
> > The 3D robustness.
>
> Information hiding seldom considers robustness. However, this also belongs to a kind of fine-tuning approach, which means the signals are embedded by fine-tuning the original model. In this situation, based on Table 2 of the main paper, such a fine-tuning strategy may show degraded 3D robustness.
>
> > Trade-off.
>
> As discussed in the **rebuttal Table2** and the previous methods [2], stronger 3D perturbations can enhance watermarking decoding accuracy, while it may affect image quality. Thus, for the envisioned scenario, there should also be a similar trade-off.
>
> However, we have to say our method is for digital watermarking, which is a different topic against information hiding, despite their somewhat correlations. We will investigate how to incorporate your suggestions into our future work and feel happy to discuss this point with you during the discussion session.
>
> > Limitations
>
> We will further incorporate more limitations and potential impacts into our final version by considering the valuable suggestions from each reviewer.
>
> [1] Simplicits: Mesh-Free, Geometry-Agnostic, Elastic Simulation.
>
> [2] StegaNeRF: Embedding Invisible Information within Neural Radiance Fields.

---

> > ### Comment · Reviewer_BKx3 · 2024-08-10
> >
> > >W1: Generalizability of decoders.
> > >>We agree that the generalization ability is very important for practical use. In our designs, we employ a two-layer protection approach to achieve generalizable and specific protection. We have two decoders in 2D and 3D, respectively. The 2D decoder is never trained per scene for generalization ability. In our setting, it is pre-trained and can be used across images rendered from different 3DGS models. This can provide protections for vast number of Gaussians distributed across the internet. Our second decoder is in 3D and is trained per scene. This is the second protection wall for 3D assets with more specific protection. The 3D asset owners can train their 3D decoder per scene based on the scene property. We believe such two-layer protection with both general and specific protection can make 3D assets safer.
> >
> > Thank you for your insightful response. It appears that we may have a slight misunderstanding regarding the concept of generalizability. While using a common decoder to fit all previously encountered scenes is certainly feasible, this approach doesn't necessarily imply true generalization. Genuine generalization refers to the applicability of a model to unseen scenarios.
> >
> > To illustrate this point, let's consider two analogous examples:
> > 1. In 2D steganography on images, the encoder and decoder should be capable of embedding and extracting watermarks on carrier images that were not seen during training.
> > 1. In machine learning, generalization typically refers to performance on unseen domains and data, rather than maintaining good performance across numerous seen domains.
> >
> > This distinction raises an important question: How well does the proposed method perform on entirely new scenes or objects that were not part of the training set that have been seen?
> > True ”generalization“ in this context would involve the ability to apply the watermarking technique to novel 3D scenes or objects without requiring retraining or significant adaptation. This capability would demonstrate the robustness and versatility of the approach, making it more applicable in real-world scenarios where encountering new and diverse 3D environments is common.

---

> > > ### Author Response · Authors · 2024-08-13
> > >
> > > Dear Reviewer BKx3,
> > >
> > > Thank you very much for your thoughtful comments and for taking the time to engage in this discussion. We greatly appreciate your insights and the opportunity to clarify our work.
> > >
> > > We will incorporate all your valuable suggestions about the generalization ability into our final version. That would be a very interesting part for the future work in this area. As the discussion has come to its end, may we know whether our response has fully addressed your concerns? If possible, we would be very grateful if you can raise your score.
> > >
> > > Thanks again for your valuable feedback during the rebuttal and discussion.
> > >
> > > Best Regards,
> > >
> > > Authors

---

> ### Author Response · Authors · 2024-08-11
>
> Thanks for pointing out this. We agree that a general encoder-decoder framework in image watermarking is indeed highly desirable. However, the unique nature of 3D neural representations presents distinctions that require a different approach when compared to image watermarking.
>
> In 3D neural representations, trainable parameters or networks are utilized to represent 3D scenes. This fundamental difference makes it challenging to directly apply the encoding techniques used in image watermarking to encode those parameters or networks. Consequently, some changes are necessary for watermarking neural representations.
>
> These changes inevitably lead to certain optimizations during the message embedding for neural representations like previous CopyRNeRF for NeRF or our approach for 3DGS. We agree that this may potentially impact the generalization ability. To address this, our current strategy focuses on minimizing the time costs of the message embedding. By reducing the time required for embedding, we can partly mitigate the issues arising from these additional optimizations.
>
> We have conducted extensive experiments here. Our results demonstrate that our method achieves significantly shorter message embedding times. However, such message embedding sometimes may need about 70 hours in established pipelines for neural representations. This improvement not only enhances efficiency but also helps to preserve the generalization capability of the watermarking technique.
>
> | Datasets  | 3DGS training |   Our message embedding	|
> | :----------| :-------------------------------: | -----: |
> | Synthetic datasets (Blender) |  30k steps / 30mins  | 1k steps / 3 mins |
> | Real-world scenes (LLFF) |  30k steps / 45mins |   1k steps / 5 mins  |
>
> Besides, we have to point out that the images used for training the 2D message decoder are all from the COCO dataset. This dataset does not have any correlations to the images used for the optimization of 3DGS models. Thus, it means that our 2D message decoder has never seen the images used for the optimization of 3DGS models, and can be used as a generalized message decoder on any novel 3D scenes.
>
> What you mention is very insightful. We will incorporate those suggestions into the future work of our final version.

---

> > ### Comment · Area_Chair_SrY1 · 2024-08-13
> >
> > @Reviewer BKx3, authors provided new comments for your questions. could you please check whether they have addressed your concerns or not. thanks!

---

> > ### Comment · Reviewer_BKx3 · 2024-08-14
> >
> > Thank you for your response. A quick follow-up question: why the training time you provide for 3DGS training: 30k steps / 30mins | 30k steps / 45mins is so big?

---

> > > ### Author Response · Authors · 2024-08-14
> > >
> > > Dear Reviewer BKx3,
> > >
> > > Our 3DGS training follows the original 3DGS paper [1], which takes 30k steps by default. In our experiments, the typical training times required for training 3DGS ranged from 30 to 45 minutes. The training time can also be found in the original 3DGS paper [1] Table. 1, and for the 30k steps model it also ranges from about 30 to 45 minutes.
> > >
> > > Users can also select 7k steps for faster training time, and the optimization can also finish faster since the 7k steps 3DGS model contains fewer parameters and can be trained faster. No matter whether the user uses a 7k or 30k steps 3DGS model, our method can all be applicable to these existing 3DGS models and be optimized within minutes for watermarking purposes.
> > >
> > > [1] 3D Gaussian Splatting for Real-Time Radiance Field Rendering.
> > >
> > > Best regards,
> > >
> > > Authors

---

> > > > ### Comment · Reviewer_BKx3 · 2024-08-14
> > > >
> > > > Thank you for your response. Overall, I still consider this method to be relatively incremental, similar to a version of CopyRNeRF [1] that replaces NeRF with Gaussian Splatting (GS). The novelty is not that clear. From my empirical understanding, GS has a weaker capacity for hiding information compared to NeRF, which makes embedding content like images more challenging than with NeRF. This is why I asked about the ablation study on information hiding capacity in my rebuttal. However, hiding string data is much less challenging. While efficiency can partially address the issue of generalizability, on the whole, I still find the work to be a bit incremental.
> > > >
> > > > [1] CopyRNeRF: Protecting the CopyRight of Neural Radiance Fields

---

> > > > > ### Author Response · Authors · 2024-08-14
> > > > >
> > > > > Dear Reviewer BKx3,
> > > > >
> > > > > Thank you for your feedback. We would like to state that our work is **different** from CopyRNeRF and we combine our work closely with the 3DGS. We would like to highlight our novelty to clarify our contributions.
> > > > >
> > > > > **1. The difference with CopyRNeRF.**
> > > > >
> > > > > > 1. Uncertainty-based message embedding.
> > > > >
> > > > > Firstly, our work originated from the uncertainty perspective for the Gaussian Splatting model to minimize perturbations on the 3DGS models for achieving invisible watermarking. While CopyRNeRF uses an additional color mlp layer to embed copyright messages into the NeRF model. Without considering the uncertainty, naively applying the existing watermarking strategies such as CopyRNeRF can cause noticeable distortions. We have conducted the experiment in the baseline of "3DGS with message"  which is similar to CopyRNeRF that encodes the message into the 3DGS model, which can result in noticeable artifacts and reduced bit accuracy, as demonstrated in the **main paper Table 1** and **rebuttal paper Figure 1**.
> > > > >
> > > > > Besides, the HiDDeN decoder mainly focuses on the decoding of information along the boundary areas. From **the left part of rebuttal Figure 2**, such boundary areas are all with high uncertainty values. This can also show the correlation between the uncertainty and the message embedding. While CopyRNeRF utilizes an additional MLP layer to embed messages into the NeRF color component, which is totally *different* from our method.
> > > > >
> > > > > > 2. Message extraction on both 2D images and 3D Gaussians.
> > > > >
> > > > > Our method enables copyright message extraction from both 2D images and 3D Gaussians. CopyRNeRF is limited to extracting messages from 2D-rendered images, as directly extracting messages from the neural networks in NeRF proves challenging. Consequently, owners of the 3D assets can only assert their ownership through these rendered 2D images, rather than from the underlying 3D neural representation itself. Our approach allows direct message extraction from 3D Gaussians. This provides model owners with a novel means of claiming ownership directly from their 3D assets, rather than relying solely on rendered images. This advancement offers a more robust and versatile approach to protecting the copyright of 3DGS model owners.
> > > > >
> > > > > > 3. Training efficiency.
> > > > >
> > > > > CopyRNeRF usually takes 70 hours for training to embed copyright messages, while our method can finished within minutes, which can ensure that our method can be generalized to novel 3D scenes and its practicality for users to easily distribute the watermarked 3DGS models online.
> > > > >
> > > > > **2. Hiding information such as images.**
> > > > >
> > > > > We also state during the rebuttal that our proposed 3D watermark can ensure the multi-view consistency in the **rebuttal paper Figure 5**, which can answer reviewer BKx3's concern about the physical challenge that our method can ensure the watermark information can be decoded from any camera angles.
> > > > >
> > > > > We also discuss the potential challenge that reviewer BKx3 raised about hiding information such as images from the 3DGS scenes. Though we have to say our method is for digital watermarking, which is a different topic against information hiding, we are very willing to discuss that extending our method into the information hiding scenarios. Based on our 3D watermark being physically multi-view consistent, our watermark method can be extended into the information hiding cases. As suggested by reviewer BKx3, in such case the decoder may need to be changed into seperated ones for decoding media such as images. The information capacity relies on both the information encoding and decoding process. For message encoding, we can adjust our uncertainty threshold to strengthen the 3D perturbations for adding enough signal for the information decoding. For message decoding, we can select a decoder that contains large model parameters for feature extraction to detect the embedded signal and decode it into images.
> > > > >
> > > > > We hope our explanation can address the concerns. We highly appreciate your suggestions, and we will incorporate your suggestions into our final paper.
> > > > >
> > > > > Best Regards,
> > > > >
> > > > > Authors

---

### Official Review · Reviewer_VdSM · 2024-07-05

**Soundness:** 2
**Presentation:** 2
**Contribution:** 2
**Rating:** 5
**Confidence:** 5

**Summary:**

3D Gaussian Splatting(3DGS) has gradually become the mainstream method for acquiring 3D assets, which has led to a demand for copyright protection of 3DGS. In this paper, a watermarking method based on uncertainty called GaussianMarker is proposed. Firstly, 3DGS is partitioned based on uncertainty, and the watermark is only added to the model parameters with high uncertainty. Subsequently, the corresponding parameters are perturbed using both 2D and 3D watermark encoders, enabling the extraction of watermark information from rendered 2D images as well as directly from 3D model parameters. Experimental results demonstrate the robustness of the proposed GaussianMarker method against 2D and 3D distortions.

**Strengths:**

1. The paper proposes a method that utilizes uncertainty to partition 3D Gaussian. By embedding watermarks specifically in the parameters with high uncertainty, the method aims to mitigate the impact on the quality of the model.

2. The paper considers the extraction of watermarks in both 2D and 3D scenarios, taking into account the robustness of watermark extraction in these two contexts.

**Weaknesses:**

1. The paper mentions that the calculation of uncertainty is related to the model parameters, and in 3D Gaussian, each point has multiple parameters such as $\mu, R, S, c, and \alpha$. It would be helpful if the authors could clarify which specific parameters are used in the proposed method. Additionally, the paper provides a formula for calculating model uncertainty, but it is unclear how the uncertainty of each Gaussian is computed and used for partitioning. The authors should provide further explanation or clarification on this matter.
2. The description of the densify function $g(\cdot)$ in the paper states that it randomly samples a new position from a distribution. According to my understanding, the original Gaussian $G_i$ should have been replaced. However, Figure 2 shows that the original Gaussian  $G_i$ still exists, which is confusing to me.
3. During the watermark embedding process, it is unclear whether the 2D and 3D watermarks are embedded into the same model parameters. It would be helpful if the authors could clarify which specific model parameters of the 3D Gaussian are used for embedding the watermarks.
4. In the section on "Distilling watermarking knowledge," the authors mention that "the pre-trained feature from 2D space can be distilled to the 3D space." It is important for the authors to provide an explanation of how this is achieved.

**Questions:**

1. In the experimental section, the authors present four baseline methods. How do 3DGS with message and 3DGS with fine-tuning extract messages.
2. Four types of 3D editing methods are listed in the experiment, which parameters of 3DGS are affected by these distortions?

**Limitations:**

The authors have addressed the limitations.

---

> ### Author Rebuttal · Authors · 2024-08-05
>
> Dear Reviewer VdSM,
>
> Thank you for your valuable feedback and constructive comments. We will address your comments below and in the revised paper.
>
> ### Weakness
>
> > W1: Parameters for uncertainty calculation and partitioning.
>
> As we have mentioned in the main paper (Section 4.1), the model parameters $\theta$ are used to compute the uncertainty, including all Gaussian parameters $\mu$, $R$, $S$, $c$, and $\alpha$. The uncertainty is estimated by calculating the Hessian matrix as an approximation of the Fisher information. This process can be simplified by computing the gradient for each Gaussian (main paper, lines 148-152) as $\mathbf{H}\left[\mathbf{I} \mid \mathbf{V}, \boldsymbol{\theta}^*\right]=\nabla_{\boldsymbol{\theta}} f\left(\mathbf{V} ; \boldsymbol{\theta}^*\right)^T \nabla_{\boldsymbol{\theta}} f\left(\mathbf{V} ; \boldsymbol{\theta}^*\right)$. Since Fisher information is additive (main paper, lines 153–155), the uncertainty of each Gaussian can be summed by adding the uncertainty of each parameter: $\mathbf{H}[\mathbf{I}|\mathbf{V},\mathbf{\theta}^*] = \mathbf{H}[\mathbf{I}|\mathbf{V},\mathbf{\theta^*_{\mu}}] + \mathbf{H}[\mathbf{I}|\mathbf{V},\mathbf{\theta^*_{R}}] + \mathbf{H}[\mathbf{I}|\mathbf{V},\mathbf{\theta^*_{S}}] + \mathbf{H}[\mathbf{I}|\mathbf{V},\mathbf{\theta^*_{c}}] + \mathbf{H}[\mathbf{I}|\mathbf{V},\mathbf{\theta^*_{\alpha}}]$. The high uncertainty 3D Gaussians are partitioned based on the Equation (7).
>
> > W2: The desify function 𝑔(⋅).
>
> As we have mentioned in the main paper (lines 163-170), we retain the integrity of the original 3D Gaussians, denoted as $\mathcal{G}$. We densify the 3D Gaussians with high uncertainty, and the newly added 3D Gaussians are our proposed 3D perturbations denoted as $\mathcal{\tilde{G}}$. We embed $\mathcal{\tilde{G}}$ into $\mathcal{G}$ to get the watermarked Gaussians: $\mathcal{\hat{G}} = \mathcal{G} \cup \mathcal{\tilde{G}}$. We keep the integrity of the original Gaussians $\mathcal{G}$ and add 3D perturbations $\mathcal{\tilde{G}}$ on the original 3D Gaussian, similar to the 2D watermarking methods apply an invisible 2D perturbation messages on the original cover images: $x_{w} = x_{o} + \delta$ (main paper, lines 178-180).
>
> > W3: It is unclear whether the 2D and 3D watermarks are embedded into the same model parameters
>
> The 2D and 3D watermarks are all embedded into the same model parameters. As mentioned in lines 207-210 of the main paper, our training contains two phases. In the first phase, we distill the messages into the model parameters. After this distillation, the messages have already been able to be extracted from 2D rendered images. In the second phase, we only train the 3D message decoder to ensure that the messages embedded in the first phase can be directly extracted from the 3D assets.
>
> > W4: Distilling watermarking knowledge.
>
> As we have mentioned in lines 211-213 of the main paper, previous methods have shown that the 2D knowledges can be distilled into the 3D radiance fields via additional settings. In our designs, we distill the 2D knowledges from a pre-trained HiDDeN decoder into the Gaussian parameters as the embedded watermarks. Based on our proposed two training phases, such distilled knowledges can be extracted from the 2D and 3D domains.
>
> ### Questions
>
> > Q1: Message extraction methods
>
> In our experimental setting, for a fair comparison, “3DGS with message” and “3DGS with fine-tuning” methods all use the same 2D and 3D message decoders to extract messages. The 2D message decoder is a pre-trained HiDDeN decoder and can extract messages on the watermarked rendered images (main paper, lines 176–177). The 3D message decoder is based on PointNet architecture and can extract messages on the watermarked 3D Gaussians (main paper, lines 196–197).
>
> > Q2. Four types of 3D editing methods are listed in the experiment, which parameters of 3DGS are affected by these distortions?
>
> The four types of 3D editing (3D Gaussian noise, translation, rotation, crop-out) occur in 3D space and are used to modify the positions $\mu$ of the 3D Gaussians, while other Gaussian parameters remain unchanged. We treat the 3D Gaussian mean $\mu$ as the point position and other parameters as the associated point features (main paper, lines 193-196), so that the PointNet-based 3D message decoder can extract the hidden messages directly from the 3D Gaussians and even they are spatially transformed or distorted.

---

> > ### Comment · Reviewer_VdSM · 2024-08-11
> > **Some questions**
> >
> > Thank you for your careful response. Combining it with the feedback from other reviewers, I still have some questions.
> >
> > 1. Does the watermark in question amount to introducing an optimizable perturbation in a 3D scene (a set of densify high-uncertainty Gaussians)? Regardless of the 2D or 3D phase, are all parameters of these Gaussians optimized ($\mu, R, S, c, and \alpha$)?、
> > 2. I am unclear on how the 2D information is distilled. From my understanding, is it rendered directly from a viewpoint, then a watermark is added to the rendered image, and the corresponding Gaussian parameters are updated using gradients. Is this correct?
> > 3. Regarding the four 3D editing methods mentioned, the authors stated that only  $\mu$ was modified. However, as per my understanding, some existing 3D Gaussian editing methods, such as Gaussianeditor[1], modify all parameters of the Gaussian. In this case, does the PointNet-based 3D message decoder proposed in this paper become ineffective?
> >
> > [1] Chen, Yiwen, et al. "Gaussianeditor: Swift and controllable 3d editing with gaussian splatting." *Proceedings of the IEEE/CVF Conference on Computer Vision and Pattern Recognition*. 2024.

---

> > > ### Author Response · Authors · 2024-08-12
> > >
> > > Thanks for your kindly reply and we are willing to further discuss about the technical details:
> > >
> > > 1. The optimizable perturbations.
> > >
> > > Yes, our proposed method introduces an optimizable perturbation $\tilde{\mathcal{G}}$, which is added into the original Gaussians $\mathcal{G}$ to obtain watermarked Gaussians $\hat{\mathcal{G}}$ used for the 3D scene representation. Those perturbations parameters in $\tilde{\mathcal{G}}$ are all optimized in the 2D phase, including $\mu$, $R$, $S$, $c$, and $\alpha$. In the 3D phase, we use the watermarked Gaussians $\hat{\mathcal{G}}$ (which contain the perturbations $\tilde{\mathcal{G}}$) optimized in the 2D phase for training the 3D message decoder.
> > >
> > > 2. The 2D watermark distilling.
> > >
> > > Yes, you are correct! The watermark is added into the rendered image. Let's review the process of the 2D watermark distillation in our proposed framework. At the beginning we only have the original Gaussians $\mathcal{G}$, and after we apply our proposed uncertainty-aware perturbation based on the Equation (6) and (7), we obtain the 3D perturbations $\tilde{\mathcal{G}}$ as our proposed 3D watermark, and the final watermarked Gaussian is denoted as $\hat{\mathcal{G}} = \mathcal{G} \cup \tilde{\mathcal{G}}$. Under the supervision of the pre-trained 2D message decoder, the corresponding Gaussians parameters in $\tilde{\mathcal{G}}$ are updated using the gradient in Equation (8). When the optimization is finished, the watermarked Gaussians parameters $\hat{\mathcal{G}}$ contain the watermarking information. Those watermarks can then be transmitted into the rendered image pixels $\hat{C}$ based on the rendering defined in Equation (2).
> > >
> > > 3. Modifying other Gaussian parameters and GaussianEditor.
> > >
> > > Thanks for your insightful suggestions. Such robustness you mentioned is indeed a very important aspect of digital watermarking. Besides, as PointNet-based message decoder is more sensitive to geometry editing, we mainly focus on the evaluations of some operations like translation, rotation and crop-out. We further conduct experiments to edit all Gaussian attributes by adding a normalized noise $n \sim \mathcal{N}(0, \sigma)$ as random perturbations and the results are shown in the table below. The results indicate that our 3D message decoder still shown robustness when other Gaussian attributes are modified:
> > >
> > >  Method  | None |   $c$  |    $R$  |    $S$  | $\alpha$| All ($\mu$, $c$, $R$, $S$, $\alpha$) |
> > > | :----------: | :------: | :----: | :------: | :------: | :------: |  :-----: |
> > > | Noise ($\sigma=0.1$) | 100% | 97.69% | 98.95%  |  98.39%  | 99.30%|   96.61% |
> > > | Noise ($\sigma=0.5$) | 100% | 97.15% | 98.94%  |  98.13%  | 99.23%|   95.82% |
> > >
> > > Based on your valuable suggestions, we have further tried the modifications in GaussianEditors on the MipNeRF360 flowers scene. We have considered the color-editing scenarios by adding prompt "make it blue" and "make it red". Similar to the dynamic scene situations in the rebuttal Figure 4 and Figure 5, since our distortion layers are scalable, we can achieve color robustness by adding color-jittering into the distortion layers during the 2D message decoder training. As shown in the table below, our method still keeps high bit accuracy even when the scene color is edited.
> > >
> > >  Method  | Scene| None |  "make it blue" | "make it red" |
> > > | :----------:| :------: |  :------: | :------: | :-----: |
> > > | GaussianEditor | Flowers | 97.91% | 93.75% | 91.66% |
> > >
> > > Achieving robustness in different modifications is very important. If the users have specific needs for different robustness, they can change the distortion laters to achieve higher robustness. However, it is also very difficult to cover all distortions at the same time. Making the watermarking system robust is an active research area. We will incorporate your valuable concerns and suggestions into our final version.

---

> > > > ### Comment · Area_Chair_SrY1 · 2024-08-13
> > > >
> > > > @Reviewer VdSM, authors provided new comments for your questions. could you please check whether they have addressed your concerns or not. thanks!

---

> > > > ### Author Response · Authors · 2024-08-13
> > > >
> > > > Dear Reviewer VdSM,
> > > >
> > > > Thanks very much for your constructive feedback throughout this review process. We greatly appreciate the time and effort you have put into evaluating our work and providing insightful comments and questions.
> > > >
> > > > We are grateful for your reviews on highlighting important technical details for clarification of our proposed methodology. We have provided further detailed descriptions and experiments on the uncertainty calculation and thresholding to provide more in-depth explanations of our methodology and results. We hope that our responses have adequately addressed your concerns.
> > > >
> > > > As the deadline for the discussion has come to its end, we would be very grateful if you can kindly raise your score. Thanks again for your valuable contributions to improving our paper. Your feedback has been extremely helpful, and we are grateful for your expertise and insights.
> > > >
> > > > Best regards,
> > > >
> > > > Authors

---

> > > > ### Comment · Reviewer_VdSM · 2024-08-14
> > > > **Response**
> > > >
> > > > The authors have addressed some of my concerns. I will raise my score.

---

> > > > > ### Author Response · Authors · 2024-08-14
> > > > >
> > > > > Dear Reviewer VdSM,
> > > > >
> > > > > We sincerely thank you for your thorough review process and for raising your score. Your insightful feedback has been invaluable in improving our work, and we deeply appreciate your time and expertise. We're grateful for your positive evaluation and will incorporate your suggestions in our final paper.
> > > > >
> > > > > Best regards,
> > > > >
> > > > > The Authors

---

### Official Review · Reviewer_QvjQ · 2024-07-10

**Soundness:** 3
**Presentation:** 3
**Contribution:** 3
**Rating:** 7
**Confidence:** 3

**Summary:**

The paper presents a new method for embedding digital watermarks in 3D Gaussian Splatting (3DGS) models to protect the copyright of 3D assets. Traditional watermarking techniques for mesh, point cloud, and implicit radiance fields are not suitable for 3DGS, as they can cause distortions in rendered images. The authors propose an uncertainty-based approach that constrains perturbations to the model parameters, ensuring that watermarks remain invisible while preserving visual quality. The method allows for reliable extraction of copyright messages from both 3D Gaussians and 2D rendered images, even under various distortions.

**Strengths:**

1. The proposed method ensures that the embedded watermarks do not cause significant distortions in the rendered 3D scenes or 2D images, maintaining the visual quality of the assets.
2. The approach is designed to be robust against various forms of 3D and 2D distortions, such as noise, translation, rotation, cropping, JPEG compression, scaling, and blurring. This enhances the reliability of copyright protection.
3. The method allows for the extraction of copyright messages from both 3D Gaussian parameters and 2D rendered images, providing multiple layers of security and verification.
4. Extensive experiments demonstrate that the method achieves state-of-the-art performance in both message decoding accuracy and view synthesis quality.

**Weaknesses:**

The malicious scenarios considered are limited to traditional distortions. \
More sophisticated scenarios should also be explored. \
For instance, a malicious actor could fine-tune the downloaded 3DGS or use an auto-encoder to remove embedded information ([1],[2],[3]). \
In such cases, how would the proposed method perform?

Additionally, a more complex scenario to consider is when a malicious actor renders Bob's 3DGS and uses it as training data to create their own 3DGS. \
How would the proposed method address these advanced threats?

[1] Fernandez et al., The Stable Signature: Rooting Watermarks in Latent Diffusion Models \
[2] Kim et al., WOUAF: Weight Modulation for User Attribution and Fingerprinting in Text-to-Image Diffusion Models \
[3] Zhao et al., Invisible Image Watermarks Are Provably Removable Using Generative AI

**Questions:**

Please refer weakness.

---

> ### Author Rebuttal · Authors · 2024-08-05
>
> Dear Reviewer QvjQ,
>
> Thank you for your valuable feedback and constructive comments. We appreciate your suggestion about considering more sophisticated scenarios, and we will address your comments below and in the revised paper.
>
> > W1: Model fine-tuning and auto-encoder attack.
>
> By following your suggestions, we have designed experiments for both the model fine-tuning scenario and the auto-encoder attack scenario.
>
> For the **model fine-tuning**, we consider a challenging case where attackers have direct access to the original non-watermarked images used for the creation of the 3DGS. Based on this assumption, we implement the attack by eliminating the message loss and then fine-tuning the model solely through perceptual loss.
>
> As shown in **Figure 3, left part of the rebuttal paper**, the bit accuracy still remains relatively high even when such fine-tuning compromises the model quality. It indicates that model fine-tuning cannot significantly reduce bit accuracy without undermining image quality.
>
> For the **autoencoder attack**, we consider two kinds of VAE attacks: VQ-VAE [1], and VQ-GAN [2]. We use different compression rates to alter image quality and evaluate the robustness of the watermarking.
>
> As shown in **Figure 3 of the rebuttal paper**, the message decoding can maintain relatively high accuracy and achieve good reconstruction quality (PSNR > 31), when the two kinds of auto-encoder use a low compression rate. Then, under a high compression rate, these two auto-encoder attacks can result in lower bit accuracy and degraded image quality. Although these two kinds of auto-encoder attacks can undermine the watermark in the rendered images, as shown in the **Figure 3, right part of the rebuttal paper**, such degraded image quality may lead to multiview inconsistency, making the sharing of the rendered contents difficult.
>
> > W2: Retrain a 3DGS with Bob's rendered images.
>
> We appreciate your insightful feedback. As shown in the table below, our experiments on retraining a new 3DGS model using Bob's rendered images reveal that, even with a slight reduction in message decoding accuracy, the fully converged retrained model can still maintain a relatively high bit accuracy.
>
> |  Method     |   PSNR   |   SSIM   |   LPIPS  |   None   |  Noise   |  JPEG |  Scaling | Blur |
> | :--------------| :---------: | :----------: | :--------: | :---------: | :---------: | :---------: | :---------: | -----: |
> |  Original    |   31.97   |  0.9098  |   0.0759  | 98.94% |  98.17% | 92.41% |  97.27% | 98.20% |
> |  Retrained |   31.09   |  0.8844  |   0.0781  | 87.21% |  86.98% | 81.35% |  83.88% | 86.76% |
>
> We appreciate all your valuable suggestions and look forward to discussing them further with you during the discussion session.
>
> [1] Neural Discrete Representation Learning.
>
> [2] Taming Transformers for High-Resolution Image Synthesis.

---

> > ### Comment · Reviewer_QvjQ · 2024-08-11
> >
> > Thank you for your response. Including this additional work in the revised version will significantly enhance the clarity and completeness of your paper, making it more robust and comprehensive. My concerns are now resolved, and I have decided to increase my score. Thank you for your efforts.

---

> > > ### Author Response · Authors · 2024-08-11
> > >
> > > Thank you for taking the time to review our revised submission. We greatly appreciate your recognition of the additional work we've incorporated and how it has enhanced the clarity and comprehensiveness of our paper.
> > >
> > > We're pleased to hear that our revisions have addressed your concerns. Your constructive comments help us improve the quality and robustness of our work.

---

> > > ### Author Response · Authors · 2024-08-13
> > >
> > > Dear Reviewer QvjQ,
> > >
> > > Thank you once again for your thorough review and valuable feedback throughout this process. We are deeply grateful for your recognition of our additional work and how it has enhanced the clarity and comprehensiveness of our paper. We appreciate the time and expertise you've dedicated to reviewing our submission.
> > >
> > > Sincerely,
> > >
> > > Authors

---

### Official Review · Reviewer_fyQG · 2024-07-14

**Soundness:** 3
**Presentation:** 3
**Contribution:** 3
**Rating:** 5
**Confidence:** 4

**Summary:**

This paper proposes an uncertainty-based method to achieve watermarking for 3D Gaussian Splatting. Specifically, the Hessian matrix is used to estimate the parameter uncertainty. Then, the 3D Gaussians with high uncertainty are densified. The densified 3D Gaussians are trained to embed watermarking using a pre-trained 2D message decoder. After that, a 3D message decoder is trained using PointNet. Experimental results show that the proposed method achieves the best performance.

**Strengths:**

1. This paper is well-written and easy to follow.

2. The experimental results show that the proposed method achieves new SOTA results.

3. The proposed method can decode watermarking both in 2D rendered images and 3D assets.

4. An uncertainty-based method is proposed to select trainable 3D Gaussians, which is reasonable.

**Weaknesses:**

1. One concern about this paper is its novelty. The major contribution of this paper is the introduction of uncertainty into 3D Gaussians watermarking. As the definition of uncertainty using Fisher Information comes from [42], simply using uncertainty for 3D Gaussians watermarking is quite simple and straightforward. Regarding the message decoders, they are all standard operations. HiDDeN [11] is used for the 2D message decoder, and PointNet [43] is used for the 3D message decoder. Therefore, the major contribution of the proposed method should be further justified.

2. The proposed method utilizes the 3D Gaussians with high uncertainty to embed watermarking. What if an attacker also uses this feature? The attacker could first identify the 3D Gaussians (after training/fine-tuning) with high uncertainty and then only attack these 3D Gaussians using techniques such as Noise, Translation, Rotation, or Cropout. Additionally, the attacker might delete some of the identified 3D Gaussians to compromise the 3DGS assets.

3. The influence of the parameter uncertainty threshold should be included in the experiments to assess the sensitivity of the uncertainty threshold on the proposed method.

4. The results with different bit lengths are missing.

**Questions:**

See Weakness.

**Limitations:**

The authors have adequately addressed the limitations.

---

> ### Author Rebuttal · Authors · 2024-08-05
>
> Dear Reviewer fyQG,
>
> We will address your comments below and in the revised paper.
>
> > W1. One concern about this paper is its novelty.
>
> Thanks for raising the concern. While our approach incorporates elements from classical techniques, our contributions extend beyond these traditional frameworks.  Besides the **first exploration** to protect the 3DGS via digital watermarking, our primary contributions also lie in developing a framework based on **uncertainty** for embedding invisible perturbations into the 3DGS model, and **message extraction from both 2D images and 3D Gaussians**.
>
> **W1-1. Use uncertainty to achieve invisible watermarking of 3DGS**
>
> One major contribution is the application of uncertainty estimation in invisible watermarking of 3DGS, an exploration that diverges from its traditional use in active learning [1]. We find that uncertainty estimation can inherently identify model parameters that are more tolerant to perturbations, making it well-suited for the purpose of invisible watermarking in 3DGS.
>
> Without considering the uncertainty, naively applying the existing watermarking strategies can cause noticeable distortions. As shown in **rebuttal Figure 1**, compared with our method, "3DGS with message" and "3DGS with fine-tuning" all show poor reconstruction quality and reduced bit accuracy. This further demonstrates the superiority of the proposed uncertainty-based strategy. Besides,  the HiDDeN decoder mainly focuses on the decoding of information along the boundary areas.  From the **left part of rebuttal Figure 2**, such boundary areas are all with high uncertainty values. This can also partly show the correlation between the uncertainty and the message embedding.
>
> **W1-2. Message extraction from both 2D images and 3D Gaussians.**
>
> Our second contribution is a method that enables copyright message extraction from both 2D images and 3D Gaussians.  Existing watermarking techniques for 3D assets (e.g., NeRF) are limited to extracting messages from 2D rendered images, as directly extracting messages from the neural networks used for scene representation in NeRF proves challenging. Consequently, owners of the 3D assets can only assert their ownership through these rendered 2D images, rather than from the underlying 3D neural representation itself. Our approach allows direct message extraction from 3D Gaussians using PointNet. This provides model owners with a novel means of claiming ownership directly from their 3D assets, rather than relying solely on rendered images. This advancement offers a more robust and versatile approach to protecting the copyright of 3DGS model owners.
>
> > W2: The proposed method utilizes the 3D Gaussians with high uncertainty to embed watermarking. What if an attacker also uses this feature?
>
> In response to your concern about this potential risk, we have conducted additional experiments specifically attacking 3D Gaussians with high uncertainty values to assess the robustness of our approach. We assume a scenario where an attacker directly manipulates the high-uncertainty 3D Gaussians. As demonstrated in the table below, our method maintains relatively high bit accuracy despite this attack.
>
> This resilience partly stems from the fact that the uncertainty distribution of watermarked 3D Gaussians can differ from that of the original 3D Gaussians, as mentioned in the main paper (lines 168–170). If the attacker conducts the operation based on the original Gaussians, it is still challenging to attack the watermarked Gaussians with new properties. Additionally, our 3D message decoder randomly samples 3D Gaussians (main paper, lines 197-199) so that even if some of the high-uncertainty Gaussians are deleted, messages can still be extracted from other 3D Gaussians. However, it is still suggested that the watermarking strategy be kept private.
>
>
>
> | Method|   Noise 	|   Translation  |   Rotation  |  Cropout | Delete |
> | :----------| :----------: | :----------------: | :------------: | :---------: | ---------: |
> | Normal attack |   99.95%  |   98.32%  |  95.32%  |   91.73%  |   N.A.  |
> | High-uncertainty attack  |  90.80% |  88.75% |  84.42%  | 81.97% |  77.93%  |
>
> > W3: The influence of the parameter uncertainty threshold should be included.
>
> In our experiments, we set the average uncertainty value as the default threshold. By following your suggestion, we show more results to verify the sensitivity of the uncertainty threshold. As shown in **right part of the rebuttal Figure 2**, a lower threshold can enhance the bit accuracy, though it slightly compromises image quality. Conversely, a higher threshold results in better image quality and a more lightweight model but also slightly compromising message decoding accuracy. However, in both situations, the compromises are moderate.
>
> > W4. The results with different bit lengths are missing.
>
> As we have discussed in the main paper (lines 273-275), we evaluate the message decoding capacity by setting the bit length to 48 bits, aligned with the maximum length used in 3D model watermarking methods [34, 4]. Shorter message bit lengths typically yield higher decoding accuracy. We evaluate the 16-bit and 32-bit message decoding accuracy on the Blender dataset and show the results in the below table.
>
> | Bit-length |   PSNR     |   SSIM   |   LPIPS  |   None   | Noise  | JPEG |  Scaling  |  Blur  |
> | :------------ | :-----------: | :----------: | :---------: | :---------: | :----------: | :----------: |  :----------: |  ----------: |
> |     16        |   32.25      |   0.9102 |  0.0758  | 99.53% |   98.66%  |  92.35%  |  97.63%  |   98.56% |
> |     32        |   31.97      |   0.9098 |  0.0759  | 98.94% |   98.17%  |  92.41%  |  97.27%  |   98.20% |
>
> [1] Unifying approaches in active learning and active sampling via fisher information and information-theoretic quantities.

---

> > ### Comment · Reviewer_fyQG · 2024-08-11
> >
> > Thanks for your response, my concerns have been addressed.

---

> > > ### Author Response · Authors · 2024-08-13
> > >
> > > Dear Reviewer fyQG,
> > >
> > > We greatly appreciate the time and effort you have invested in evaluating our work and providing insightful comments and questions.
> > >
> > > We want to express our thanks again for your expertise and insights. Your feedback has played a crucial role in improving our paper, and we are sincerely grateful for your dedication to the review process.
> > >
> > > Best regards,
> > >
> > > Authors

---

> ### Author Response · Authors · 2024-08-12
>
> We are pleased to hear that our revisions have addressed your concerns. Your constructive comments help us improve the quality and robustness of our work.
>
> We will incorporate all your valuable suggestions into our final versions.  If you have further questions, please feel free to raise them.  We would be grateful if you could consider raising your scores. Thanks very much.
>
> Best Regards,
>
> Authors

---

### Author Rebuttal · Authors · 2024-08-06

Dear Reviewers,

We sincerely thank all reviewers for their comprehensive evaluations and valuable feedback. We are pleased to address any additional questions during the discussion period.

Best Regards,

Authors of Paper 3674

---

### Comment · Area_Chair_SrY1 · 2024-08-11

Dear Reviewers,

Authors have provided their rebuttal. Could you please take a look to see whether your concerns are addressed or not? Thanks!

---

### Decision · Program_Chairs · 2024-09-25

**Decision:**

Accept (poster)

**Comment:**

To make this decision, AC carefully read the paper, reviewers' comments and author's rebuttal.

Before rebuttal this paper got 1 borderline accept, 1 weak accept, 2 borderline rejects. After rebuttal, 2 of the reviewers raised their score, so the final rating for this paper is 1 accept, 2 borderline accepts and 1 borderline rejects.

Before rebuttal, Reviewer fyQG raised concerns on novelty of this paper and asked for more experiment. Authors added more experiments in the rebuttal and clarified their contribution. After rebuttal Reviewer fyQG commented that authors have addressed their concerns.

Before rebuttal, Reviewer QvjQ asked for more experiments. Authors added experiments in the rebuttal and Reviewer QvjQ was happy with that and raised the rating from weak accept to accept.

Before rebuttal, Reviewer VdSM asked for some clarification and design details. After rebuttal, Reviewer VdSM said that authors have address their concerns and raised rating from borderline reject to borderline accept.

Before rebuttal, Reviewer BKx3 raised concerns on novelty, the decoder is trained per scene, rather than being a generalizable decoder, need more experiments on complicated scenes, etc. AC also echoed that the decoder is trained per scene might be a problem. But as authors explained that 2D decoder is not per scene, only the 3D decoder is per scene, which is acceptable. Reviewer BKx3 also raised concerns that only do bit message embedding probably not enough. But given bit message embedding is enough for the copyright projection which is an important usage for wateramarking, AC think it is acceptable. At the end Reviewer BKx3 kept borderline reject rating.

Considering all these, AC made the accept decision.